# Revealing the relationship between liquid fragility and medium-range order in silicate glasses

Ying Shi [1] ✉, Binghui Deng [1], Ozgur Gulbiten [1], Mathieu Bauchy [2], Qi Zhou[2], Jörg Neuefeind [3], Stephen R. Elliott[4], Nicholas J. Smith [1] & Douglas C. Allan [1]

Despite decades of studies, the nature of the glass transition remains elusive. In particular, the sharpness of the dynamical arrest of a melt at the glass transition is captured by its fragility. Here, we reveal that fragility is governed by the medium-range order structure. Based on neutron-diffraction data for a series of aluminosilicate glasses, we propose a measurable structural parameter that features a strong inverse correlation with fragility, namely, the average medium-range distance (*MRD*). We use in-situ high-temperature neutron-scattering data to discuss the physical origin of this correlation. We argue that glasses exhibiting low *MRD* values present an excess of small network rings. Such rings are unstable and deform more readily with changes in temperature, which tends to increase fragility. These results reveal that the sharpness of the dynamical arrest experienced by a silicate glass at the glass transition is surprisingly encoded into the stability of rings in its network.

Glasses with different compositions can display different rates of dynamical arrest at the glass transition, which are captured by the curvatures in their viscosity-temperature relationships. Revealing whether this dynamical arrest is encoded in the atomic structure of glass-forming systems would be of critical importance, since it not only governs all stages of industrial glass production but also could shed new light on fundamental principles governing glass transition and relaxation phenomena[1]. This is one of the most challenging topics in condensed-matter physics[2].

Near the glass transition, the viscosity-temperature dependence of glass-forming liquids can be well represented by two parameters: the glass transition temperature ($T_g$) and the fragility index ($m$)[3]. Based on the pioneering works of Oldekop[4], Laughlin and Ulhmann[5], and Angell[6], the viscosity curve can be presented in the Oldekop-Laughlin-Uhlmann-Angell (OLUA) format by plotting the base-10 logarithm of the viscosity ($\log_{10}\eta$) against the inverse temperature, scaled by the glass transition temperature ($T_g/T$), with temperatures expressed in Kelvin. For any composition, the glass transition temperature is defined as the temperature at which the equilibrium shear viscosity is equal to some fixed value, such as $10^{12}$ Pa·s, following the Angell convention[7]. If one assumes that all silicate liquids exhibit a universal viscosity limit at infinite temperature ($\eta_\infty$) of $10^{-2.93}$ Pa s[8], their very different viscosity curves in the $\log_{10}\eta$ vs. $T_g/T$ format can be presented in a single universal plot with the same starting and ending $\log_{10}\eta$ values, i.e., −2.93 and 12, respectively, in the $T_g$-scaled inverse-temperature range between 0 and 1, corresponding to $T_g/T_\infty$ and $T_g/T_g$, respectively. The distinct nature of the viscosity curves of different glasses can then be primarily described by the convex curvature of the $\log_{10}\eta$ vs. $T_g/T$ plot. This curvature can be further quantified by a single parameter: the kinetic fragility index ($m$), defined by Angell[6] as the rate at which the viscosity changes with temperature at $T_g$, given by:

$$m = \frac{\partial \log_{10}\eta}{\partial(T_g/T)}\Big|_{T=T_g} \qquad (1)$$

[1]Science and Technology Division, Corning Incorporated, Corning, NY 14831, USA. [2]Physics of AmoRphous and Inorganic Solids Laboratory (PARISlab), Department of Civil and Environmental Engineering, University of California, Los Angeles, CA 90095, USA. [3]Neutron Scattering Division, Spallation Neutron Source, Oak Ridge National Laboratory, Oak Ridge, TN 37831, USA. [4]Physical and Theoretical Chemistry Laboratory, University of Oxford, Oxford OX1 3QZ, UK. ✉ e-mail: shiy3@corning.com

Fragility is considered to be a kinetic property since it is defined based on the temperature dependence of the viscosity. Angell further demonstrated its thermodynamic origin by drawing a direct connection between fragility and the specific heat-capacity jump at $T_g$ ($\Delta C_P$)[6]. Therefore, the fragility index can be measured by both viscometry/dilatometry and calorimetry techniques[3]. Both measurements are labor-intensive, time consuming, and are often plagued by a high level of systematic error caused by sample preparation, crystallization tendency, high-temperature volatilization, as well as discrepancies among different data-fitting algorithms. Revealing the structural origin of fragility is fundamentally important to glass science, and could help enable better prediction of these critical material parameters free from such cumbersome experiments.

We now explore which aspect(s) of the atomic structure could serve as a predictive parameter that is indicative of the sharpness of the glass transition. Fragility characterizes how rapidly a supercooled liquid undergoes dynamical arrest in response to a temperature decrease near $T_g$. Therefore, the putative structural parameter should reflect the temperature dependence of the structural-deformation capability at $T_g$. Currently, there are two approaches to interpret the fragility of network-forming glass melts from their network structure: the temperature-dependent constraint theory (TDCT) advanced by Gupta and Mauro[9,10]; and the coarse-grained model (CGM) by Sidebottom[11,12]. Both approaches are based on the same physical premise that the temperature dependence of viscosity at $T_g$ originates from the excess entropy of structural deformation, as described by the Adam-Gibbs model[13]. However, they differ in the ways that they define and quantify the temperature dependence of structural deformation—specifically regarding the relevant length-scale of the structural deformation and its underlying mechanism.

In detail, TDCT treats the glass structure at the atomic level by enumerating both the two-body linear bond-stretching and three-body angular bond-bending constraints acting in the atomic network. An onset temperature is then assigned for each type of constraint. By the TDCT definition, such a constraint is only active when the temperature is below its onset temperature. Therefore, in TDCT, the temperature dependence of viscosity upon heating is determined by a constraint-deactivation mechanism. The number of constraints per atom ($n_c$) as a function of temperature can then serve to predict the $T_g$ and fragility of glass-forming melts with varying compositions. A caveat of the TDCT approach is the need for a definition of onset temperatures, which are often derived so as to achieve a best match between modeling and experimental data, and often act as empirical fitting parameters.

In contrast, the coarse-grained model treats the network-forming oxide polytopes as rigid units hinged together by weak A−O−A linkages (wherein A are the network-forming cationic species). By treating the rigid units as nodes, the network structure is then transformed, i.e., coarse-grained, into a network of interconnected rigid-unit nodes. This is based on the idea that, just above $T_g$, it is the weak A−O−A linkages between rigid units that first deform to enable viscous flow, while the bonds and bond angles within the rigid units remain largely undeformed. The CGM approach then correlates the fragility with the average effective coordination number for the corresponding coarse-grained network, that is, the average number of weak linkages around the rigid-unit nodes. This is expressed as a coarse-grained connectivity, $\langle n_{CG} \rangle$. For silicate glasses, it can be calculated as the average number of bridging-oxygen atoms per rigid-unit polyhedron (e.g., $SiO_4$ unit). Fragility is then found to be inversely correlated with $\langle n_{CG} \rangle$ in under-constrained (floppy) glasses. The rationale behind this inverse correlation is that, with increasing $\langle n_{CG} \rangle$ values (i.e., upon increasing the coarse-grained connectivity), more energy is required to enable deformations when a glass is heated through $T_g$, which eventually leads to lower fragility

values[11]. We agree with the conclusion of CGM that more connected (or polymerized) glass-forming liquids tend to be 'stronger', that is, to exhibit lower fragility values. We also support the CGM view that only the weakest links participate predominantly in the structural deformation at $T_g$. This idea is also not fundamentally in contradiction with TDCT since the stronger constraints acting within the rigid units (e.g., Si−O bond-stretching and O−Si−O bond-bending constraints) tend to exhibit high onset temperatures (i.e., notably higher than $T_g$) and, hence, their deactivation rate may not be relevant at temperatures close to $T_g$.

Both TDCT and CGM approaches are able to be used to derive universal interpretations for the composition dependence of fragility for a wide range of glass-forming liquids, including chalcogenide (e.g. $Ge_xSe_{1-x}$)[9,11] alkali-borate[10,11] borophosphate[14], borosilicate[15], phosphate[11,16] alkali-silicate[12], and alkali-germanate[12] glasses. Here, we do not intend to propose a third universal structural descriptor for fragility that would be generic to all types of glasses. Moreover, we doubt the existence of a universal rule that is applicable for both chalcogenide and oxide glasses. For example, chalcogenide glasses can possess diversity in dimensionality[17], such as local 2D-layer structures with extensive van der Waals-like bonding[18] or 3D-network structures[19]. Most oxide glasses only contain 3D-network structures, although boron-rich oxide glasses also can present local 2D structures due to boroxol rings[20]. Thus, glasses with very different structural types should exhibit different structural-deformation mechanisms. Here, due to their industrial relevance, we focus on aluminosilicate glasses and examine experimental data to infer the existence of a meaningful structural parameter that is convincingly correlated with fragility.

## Results

### Fragility of 48 calcium aluminosilicate glasses

We choose to focus first on the calcium aluminosilicate (CAS) system due to the large amount of published fragility data for compositions that cover a substantial portion of the ternary phase diagram[21–25] as shown in Fig. 1a. Such extensive fragility studies are driven by the fact that the CAS system is an archetypal model for alkali-free glasses used in display applications[26], and that one of the most critical performance properties of display glasses, i.e., structural relaxation, is governed by fragility. Detailed information about the compositions and fragility values is listed in Supplementary Table 1. We find a simple, but unexpected, inverse correlation between fragility index and $SiO_2$ mol% values for 48 CAS glasses (see Fig. 1b).

We speculate that, here, the $SiO_2$ mol% content just acts as a surrogate parameter, which, in turn, is covariate with a more fundamental structural parameter. As illustrated in Supplementary Note 1, we analyzed a series of room-temperature (RT) neutron total-scattering data for 27 CAS glasses to seek such a structural parameter. Specifically, we use the recently developed[27] and validated[28] RingFSDP method to characterize the medium-range-order structure of these glasses (see "Methods"). The reciprocal-space first sharp diffraction peak (FSDP) can be deconvolved into three modified Gaussian peaks with fixed positions, which correspond to the real-space distances of 3.15, 3.70, and 4.30 Å, respectively. Each peak is ascribed to a certain family of rings: large rings (≥6-membered) centered at low $Q$; medium rings (5-membered) centered at intermediate $Q$; and small rings (≤4-membered) centered at large $Q$. The integrated area of each peak is proportional to the absolute number of such specified-size rings. The relative ring-size distribution ($f_n$) is calculated from the ratio of the integrated peak area ($I_{S_{n(Q)}}$) to the total FSDP area ($I_{S_{FSDP(Q)}}$) using

$$f_n = \frac{I_{S_{n(Q)}}}{I_{S_{FSDP(Q)}}} \qquad (2)$$

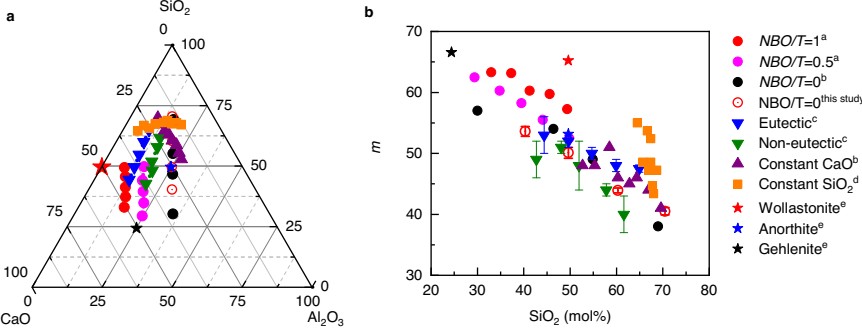

**Fig. 1 | Inverse correlation between $m$ and SiO$_2$ mol% of Group I's 48 CAS glasses. a** Compositions of 44 CAS glasses with published fragility values and 4 glasses with fragility index measured in this study, as shown in the ternary phase diagram in mol%. **b** An inverse correlation between $m$ and SiO$_2$ mol% of 48 CAS glasses, with their sources noted in the legend. The glasses in each categorized group are plotted by the same color symbols. [a]: ref. [22], [b]: ref. [21], [c]: ref. [25], [d]: ref. [24], [e]: ref. [23].

Then, the average medium-range distance (*MRD*) can be calculated as

$$MRD\,(\text{Å}) = f_{\leq 4\text{ring}} \times 3.15 + f_{5\text{ring}} \times 3.70 + f_{\geq 6\text{ring}} \times 4.30 \qquad (3)$$

The *MRD* structural parameter, which represents an average medium-range distance of the glass structure, can also be treated as a length-weighted ring-size index. The smaller the value of *MRD*, the greater the number of small-size rings. It is reciprocally related to the average position of the FSDP.

All the XRF-analyzed compositions of these glasses are listed in Supplementary Table 2. As shown in Fig. 2a, we observe the existence of a linear correlation between the calculated *MRD* and SiO$_2$ mol% values for 27 neutron-measured CAS glasses with a coefficient of determination, $R^2$ value, of 0.97 for the linear fit.

Owing to the strong linear equation for the *MRD*-SiO$_2$ mol% relationship derived from the 27 CAS neutron-scattering data, we are able to apply the fit equation shown in Fig. 2a to infer *MRD* values from SiO$_2$ mol% values for the broader set of 48 CAS glasses. The calculated *MRD* values are then correlated with their reported fragility values. However, it should be noted that the *MRD*-SiO$_2$ mol% linear equation is only applicable to the wide composition range of the CAS-glass system. Indeed, our neutron-scattering data show that no such correlation exists for the wide range of the sodium aluminosilicate (NAS) glass system. As demonstrated in the section below, NAS glasses presenting the same SiO$_2$ mol% nevertheless exhibit different *MRD* values. In such a case, the *MRD* value of glass can only be derived from the measured neutron-scattering pattern, and not inferred in a straightforward manner from its composition.

The *MRD* values for 48 CAS glasses with known $m$-values were calculated from their SiO$_2$ mol% values using the linear equation derived in Fig. 2 and listed in Supplementary Table 1. The $m$-values were then plotted as a function of their calculated *MRD* values, as shown in Fig. 2b. An inverse near-linear $m$-*MRD* correlation can be obtained by least-squares fitting of all 48 data points, with a coefficient of determination $R^2$ value of 0.65. This relatively low $R^2$ value indicates that other structural parameters, especially the *NBO/T* ratio −defined as the number of non-bridging oxygen (*NBO*) per glass-former tetrahedron (*T*)−must also play a role in determining $m$, as proposed by the coarse-grained model[12]. This influence of *NBO* is further demonstrated by the constant-SiO$_2$ data shown as the orange squares in Fig. 2b, which have varying *NBO/T* ratios and are scattered around the linear fitting line. Another reason for the scatter of data in the inverse linear correlation plot could be the uncertainties in both *MRD* and $m$-values. First, it should be noted that the compositions of the 27 neutron-analyzed CAS glasses (as shown in Supplementary Fig. 1a) only partially overlap with those of the 48 CAS glasses with known $m$-values (Fig. 1a); therefore, the calculated *MRD* values might not be accurate for some of the 48 glasses for the compositions that are outside of the range of the 27 neutron-measured glasses, where the validity of *MRD*-SiO$_2$ mol% linear correlation is not confirmed. Second, as specified in Supplementary Table 1, the $m$-values of the 48 glasses were collected from six different studies (including this one), which were measured by different methods and fitted by different equations. Systematic discrepancies between the different data sets are therefore unavoidable. Despite the relatively low $R^2$ value, the apparent inverse correlation still clearly demonstrates that the *MRD* structural parameter exhibits a strong (near-linear) correlation with fragility, over a wide range of fragility values (ranging from 37 to 67).

## Fragility of 20 MgO/CaO sodium aluminosilicate glasses

As listed in Supplementary Table 3, group II contains two sets of Na$_2$O−Al$_2$O$_3$−SiO$_2$ (sodium aluminosilicate−NAS) glasses with additions of either MgO or CaO. As shown in the ternary diagram of Fig. 3a, both sets of glasses contain constant amounts of glass modifier, substituting Al$_2$O$_3$ for SiO$_2$ along the molar composition series: (76-$x$)SiO$_2$−$x$Al$_2$O$_3$−16Na$_2$O−8RO, with $x$ = 0, 2.7,…,21.3 and 24, and R = Mg or Ca. The viscosity curves were measured by a combination of viscometry and dilatometry methods[29]. Eight of those glass-forming liquids have two $m$-values, with one being derived by MYEGA fitting[29], and the other by the modified elastic model[30]. As the accuracy of either method for assessing the 'true' fragility index cannot be independently verified, here we simplify this problem by calculating an average $m$ value based on two methods, taking the difference between the values yielded by each method as an estimation of the uncertainty. The remaining 12 glasses only have one $m$ value reported, obtained by MYEGA fitting[29]. The $m$-values of Ca and Mg-glasses are plotted as a function of Al$_2$O$_3$ mol% in Fig. 3b. Two features can be summarized from the $m$-variation trends: first, the Ca-glass-forming liquids generally have higher $m$-values than their Mg-glass counterparts; second, for the Ca-glass-forming liquids, $m$ appears to be independent (within the uncertainty) of the Al$_2$O$_3$ mol% content (except for the two highest Al$_2$O$_3$-containing glasses), while, for the Mg-glasses, $m$ increases with increasing Al$_2$O$_3$ content. This is in line with the inverse trends of *MRD* versus composition shown in Fig. 3e, obtained from the $F(Q)$-FSDPs plotted in Fig. 3c, d for Mg-NAS and Ca-NAS glasses. Overall, Ca-glasses have higher-$Q$ FSDP positions than those of Mg-glasses, corresponding to smaller *MRD* values, which are inversely correlated to the higher $m$ values. The FSDPs of Ca-glasses do not show significant changes, whereas the FSDPs of Mg-glasses systematically shift to the higher-$Q$ region as the Al$_2$O$_3$ mol% content increases, indicated by the arrow sign.

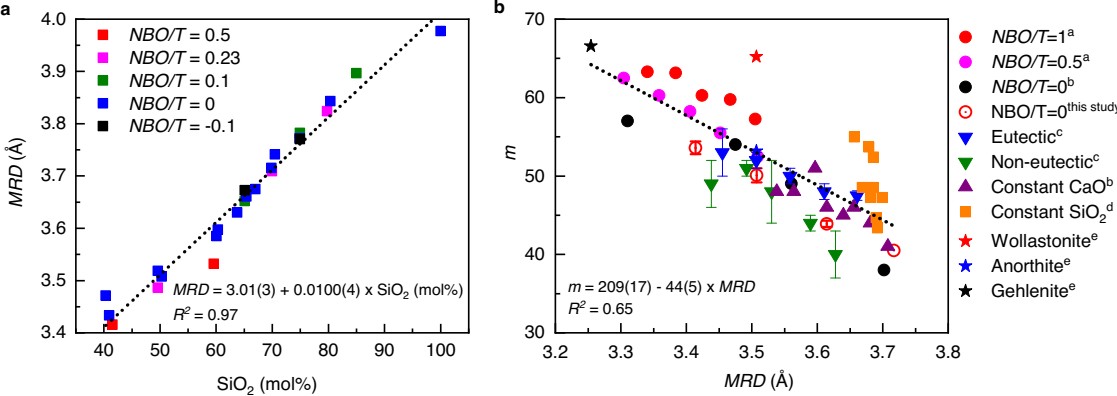

**Fig. 2 | Linear correlation between *MRD* and SiO₂ mol% and inverse correlation between fragility index (*m*) and *MRD*.** In (**a**), 27 neutron-measured glasses with the same values of *NBO/T* ratio (defined and listed in Supplementary Note 1 and Table 2) are plotted with the same color. Glasses with similar SiO₂ mol% values but different *NBO/T* ratios are shown as clustered symbols with different colors, manifesting their similar *MRD*. The linear equation is derived by least-squares fitting of the data for all glasses. The errors of both *MRD* (± 0.01 Å) and SiO₂ mol% (±0.1) values are smaller than the symbol size. In (**b**), *MRD* values of 48 CAS glasses are calculated by the linear equation obtained in (**a**). The glasses in each categorized group are plotted with the same color symbols, with their references noted in the legend. The linear equation is derived by least-squares fitting of the data for all the glasses. Only 10 glass-forming liquids (eutectic and non-eutectic glasses shown in blue and green triangles) from ref. [25] have their fragility-index errors reported; the fragility-index errors for four glasses measured in this study (red open circles) are from linear fittings, and two of those were measured twice with independent sample preparations. The error of calculated *MRD* values is 0.03 Å. ᵃ: ref. [22], ᵇ: ref. [21], ᶜ: ref. [25], ᵈ: ref. [24], ᵉ: ref. [23].

## Fragility of 13 sodium aluminosilicate glasses

Room-temperature (RT) neutron total-scattering patterns were collected for 5 NAS glasses, with their XRF-analyzed compositions and RingFSDP analysis results listed in Supplementary Table 4. As listed in Supplementary Table 5, group III contains 14 ternary NAS glasses. Here, we focus on the 13 NAS glasses with the fragility data reported in ref. 24 but not measured by neutron total scattering. As shown in Fig. 4a, the compositions of both the 13 NAS (blue) and 9 CAS (red) glasses, shown as solid circles, are scattered around the horizontal line for 67 mol% SiO₂. The $m_{vis}$-values were converted from the reported $m_{DSC}$-values[24] and are plotted as a function of *NBO/T* ratio in Fig. 4b. Glasses with 65 and 70 mol% SiO₂, which were characterized by neutron diffraction, were chosen from each system (NAS and CAS), and their compositions are shown as the solid (65 mol% SiO₂) and crossed (70 mol% SiO₂) circles in Fig. 4a. The $F(Q)$-FSDPs of these glasses are plotted in Fig. 4c, d for NAS and CAS, respectively, with the FSDPs of 70 mol% SiO₂ glasses shifted up by 0.5 for clarity. For the NAS glasses, the FSDP position shifts with different *NBO/T*-values, while the CAS-glass FSDP positions remain constant, regardless of the *NBO/T*-values. As of now, we are unable to explain the origin of this difference between NAS and CAS systems. For each system, *MRD* values are derived from the FSDP, and results are plotted as a function of *NBO/T* ratio in Fig. 4e. Linear trends of *MRD* vs. *NBO/T* ratio are assumed for both the 65 (solid blue) and 70 (dotted blue) mol% SiO₂ NAS glasses, as shown in Fig. 4e. The fitted line was subsequently used to estimate the *MRD* values of the *m*-measured NAS glasses, based on the assumption that the linear correlation between *MRD* and *NBO/T* ratio is valid in the narrow composition range of the 10 NAS glasses reported (all at constant SiO₂ mol% content). This is admittedly a rough estimation to obtain approximate *MRD* values of NAS glasses for which neutron-scattering data were unavailable. Nevertheless, the inverse *m-MRD* correlation is qualitatively demonstrated by comparing the changing trends of *m* in Fig. 4b with the changing trends of *MRD* in Fig. 4e. The higher *m*-values of CAS glasses in Fig. 4b are associated with their lower *MRD* values in Fig. 4e. Moreover, the larger *m*-slope of NAS in Fig. 4b is in line with the larger negative *MRD*-slope in Fig. 4e.

## Fragility of four industrially relevant silicate glasses

Four industrially relevant silicate glasses were studied by in-situ high-temperature neutron total-scattering analyses[31]. Their compositions, *m* values, and RT-*MRD* values are listed in Supplementary Table 6. In addition to those, we benchmark against a fictitious ideal, super-strong glass, that exhibits perfectly Arrhenius behavior for the $\log_{10}\eta$ vs. $T_g/T$ viscosity curve. In the Angell plot, it is obtained by connecting a straight line between the limiting values $\log_{10}\eta_{T_\infty} = -2.93$ and $\log_{10}\eta_{T_g} = 12$ in the $T_g$-scaled inverse temperature range between 0 and 1, corresponding to the temperature ratios $T_g/T_\infty$ and $T_g/T_g$. By definition, this fictitious glass, hereafter noted as Fictitious$_{low}$, would be a perfectly strong glass-forming liquid, and would exhibit the lowest possible fragility index, i.e., $m = 14.93$. This ideal glass is presumed to contain solely large-size rings (≥6-membered) since MD simulations reveal that ≥6-membered rings are flexible and stress-free with no internal driving force for structural deformation[32], whereas, in turn, small rings are comparatively strained and thus can exhibit some internal stress[33]. Such a glass comprising only large-size rings (≥6-membered) would have an *MRD* value of 4.30 Å, according to our RingFSDP method[27].

## Fragility (*m*)-*MRD* correlation for 89 aluminosilicate glasses

Finally, an inverse linear correlation between *m* and *MRD* can be established for all the 89 silicate glasses (Fig. 5). The *MRD* values of the 48 CAS glasses (red circles) were calculated by the linear *MRD*-SiO₂ mol% equation obtained by fitting of the data in Fig. 2a. The *MRD* values of 10 NAS glasses (black circles) were calculated from the equation obtained from fitting the data in Fig. 4e. The *MRD* values for all other glasses were obtained from the measured FSDPs of neutron structure factors. The error bars of the calculated *MRD* values were propagated from the error of the linear equation fitting. The error bars of measured *MRD* values are approximately ±0.01 Å, roughly the same as the size of symbols in the figure. The dotted line is a linear fit to the data of the 89 glass-forming liquids over a wide range of *m*-values, from 15 to 67, with an improved coefficient of $R^2 = 0.77$. This demonstrates that the inverse linear *m-MRD* correlation is applicable for silicate glasses, which, in turn, implies that fragility is strongly tied to the medium-range structural order, as extracted from a RingFSDP analysis. In Supplementary Note 2, we further confirm that such inverse linear correlation only exists between *m-MRD*, not between *m*-SiO₂ mol%.

We now propose another fictitious, super-fragile, glass-forming liquid that has the highest limit of fragility for silicate-glass materials,

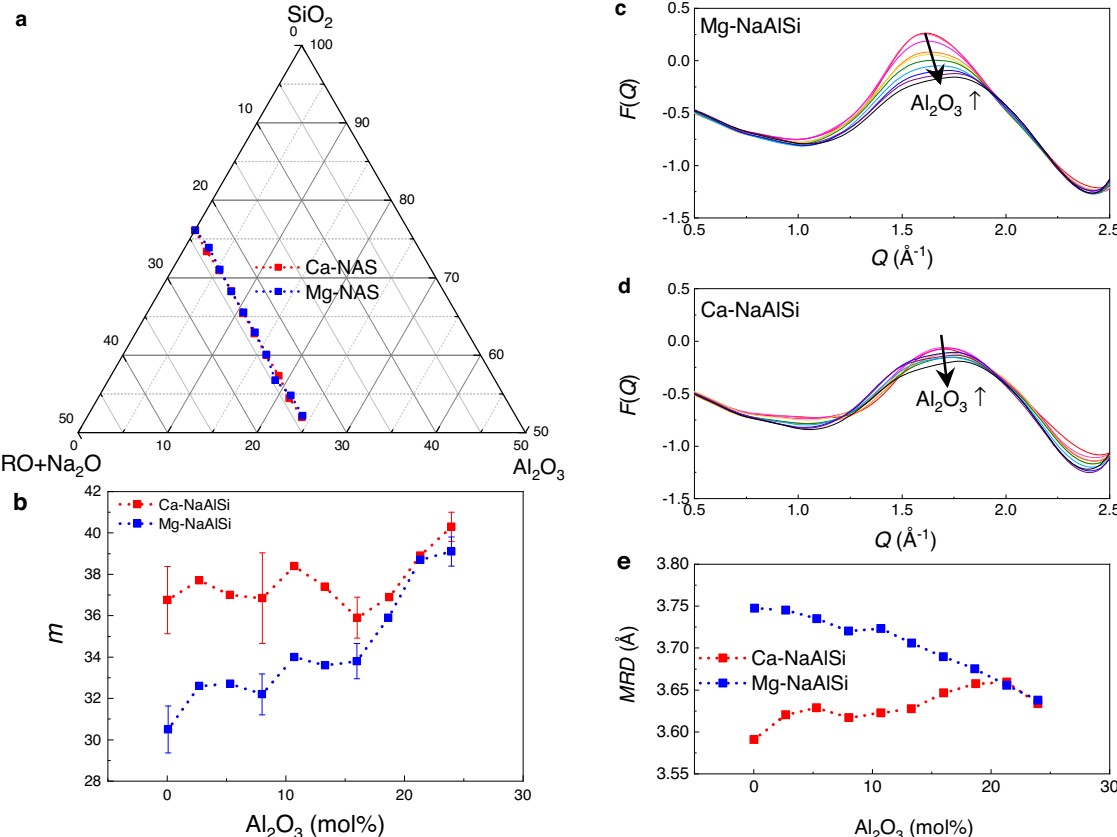

**Fig. 3 | Inverse correlation of *m-MRD* for Group II 20 MgO/CaO NAS glasses.** **a** Compositions of Ca-NAS (red) and Mg-NAS (blue) glasses, as shown in the ternary phase diagram in mol%. **b** Fragility values (*m*) as a function of $Al_2O_3$ mol% for ten Ca-NAS (red) and ten Mg-NAS (blue) glasses. The changing trends of *m* are in line with the changing trends of *MRD* (**e**) derived from the room-temperature *F(Q)*-FSDPs for Mg-NAS (**c**) and Ca-NAS (**d**) glasses. In (**c**) and (**d**), the arrow indicates the direction of $Al_2O_3$ mol% content increase; red color pattern is for non-$Al_2O_3$ containing glass, while the black color pattern is for the highest 24 mol% $Al_2O_3$ glass. Ca-glasses generally have higher *m* values than do Mg-glasses; the *m* values of Ca-glasses are independent of the $Al_2O_3$ mol%, while *m* values for Mg-glasses increase with increasing $Al_2O_3$ content. Accordingly, Ca-glasses show overall lower *MRD* values than those of Mg-glasses; no significant difference is shown for Ca-glass *MRD*s, in contrast to the *MRD* values of Mg-glasses, which decrease systematically with increasing $Al_2O_3$ mol% content.

hereafter denoted as Fictitious$_{high}$. It is assumed that this hypothetical glass should contain only small-size rings, which are highly stressed, and therefore can deform the most at $T_g$. Therefore, the Fictitious$_{high}$ glass should have an *MRD* value of 3.15 Å. Using the linear equation obtained from Fig. 5, the *m* value of the Fictitious$_{high}$ glass-forming liquid should therefore be 70. The Fictitious$_{high}$ glass estimated data-point is added to the linear correlation as the red open star in Fig. 5. It is noted that this upper bound fragility value of 70 is predicted only for silicate glasses with a ring structure; glasses with other structural types, such as molecular materials and polymers, may have much higher fragility values, with an upper limit predicted to be 180[34].

**Structural origin of *m-MRD* correlation by in situ neutron**
We emphasize that the *m-MRD* relationship revealed herein is unexpected and intriguing, since there is a priori no obvious reason for the *MRD* value (a purely static, structural parameter measured in the glassy state) to correlate with the fragility (a kinetic property measured in the supercooled liquid state). To search for the underlying structural origin of the inverse *m-MRD* relationship, we conducted in-situ high-temperature neutron total-scattering analyses with particular focus on the near-$T_g$ range[31]. Three charge-balanced CAS glasses $xCaO-xAl_2O_3-(1-2x)SiO_2$ ($x = 0.15$, 0.25, 0.3) (see Supplementary Table 7, where the nomenclature of the CAS glasses is given by the $SiO_2$ content, i.e., the glass with $x = 0.15$ is designated as CAS70) were measured at temperatures from RT to 1.1 $T_g$, and one binary sodium silicate (NS) glass $20Na_2O-80SiO_2$ (NS20) from RT to 1.25 $T_g$. The

fragility-index parameters of three CAS glasses were determined by isothermal equilibrium viscosity measurements using a three-point beam-bending method[35] around the glass transition range. Their fragility-index values are listed in Supplementary Table 7, with the detailed information presented in Supplementary Note 3. The fragility-index data of NS20 is from ref. [36].

The FSDP regions of *F(Q)* of four glasses as a function of temperature are presented in Fig. 6a. All four glasses show the FSDP position shifting to the low-*Q* side as the temperature increases, with their increasing *MRD* values summarized in Supplementary Table 8. An analog OLUA plot of scaled-$MRD_T/MRD_{T_g}$ values are plotted as a function of the scaled temperatures, $T/T_g$ (Fig. 6b). The slopes for the *MRD* change in both supercooled-liquid and glass ranges are determined by linear fits, as shown by the dotted lines. For three CAS glasses, significant slope changes are evident on passing through $T_g$. We find the existence of an inverse linear correlation between the slope ratio (liquid/glass) of *MRD* as a function of $T/T_g$ and the *MRD* value at room temperature, as plotted in Fig. 6c. We then demonstrate another positive correlation between the through-$T_g$ slope ratio of *MRD* and *m* in Fig. 6d. This qualitative correlation, based on the micro-scale liquid and glass coefficient of thermal-expansion (CTE) change, originates from the relationship discovered between viscous dynamics and the bulk configurational CTE ($\alpha_{conf}$) of silicate glasses[37]. The latter is defined as the difference between macro-scale bulk liquid and glass CTE values, $\alpha_{conf} = \alpha_{liq} - \alpha_{glass}$. The correlation observed in Fig. 6d illustrates the fact that strong glass-forming liquids show a small

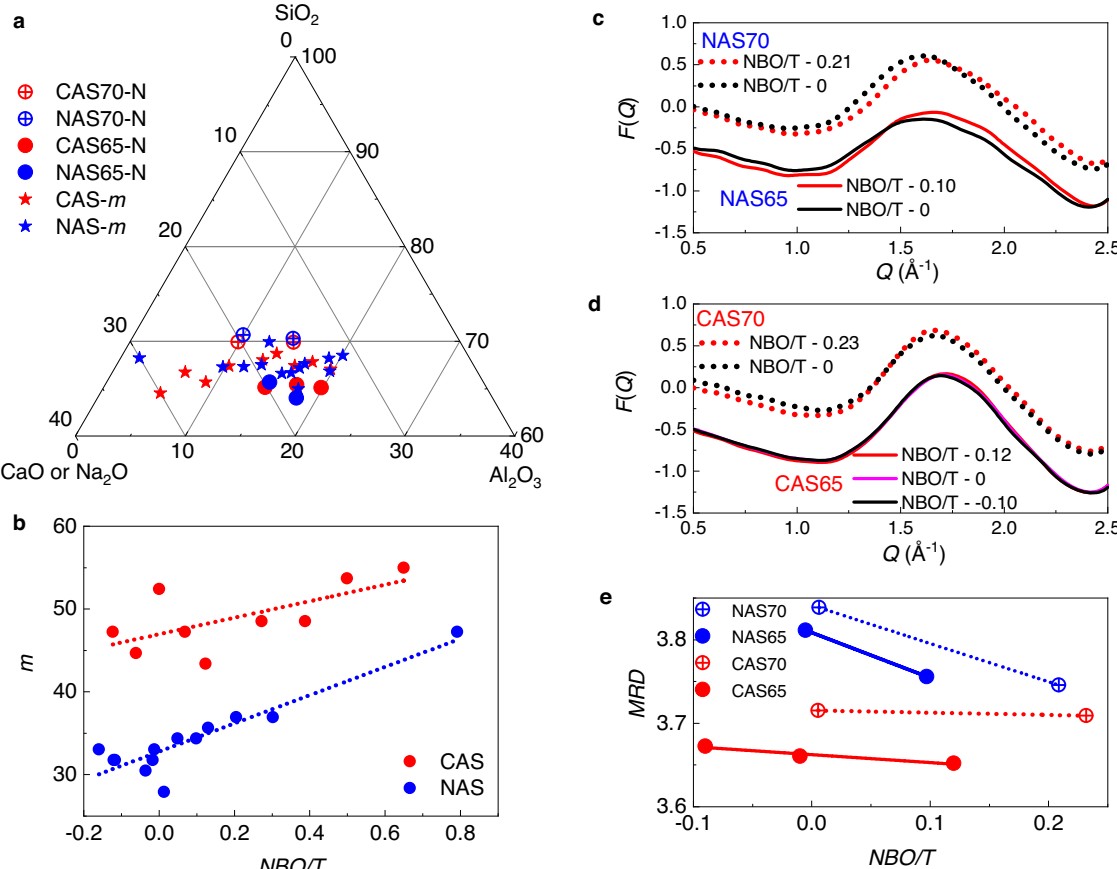

**Fig. 4 | Inverse correlation of *m*-*MRD* for Group III's 13-NAS and 9-CAS glasses.**
**a** Compositions of CAS (red) and NAS (blue) glasses, with *m* measured by a DSC
method[24] (solid symbols) and *MRD* measured by neutron total-scattering (crossed
symbols), as shown in the ternary phase diagram in mol%. **b** Fragility index (*m*)
is plotted as a function of *NBO/T* ratio for CAS (red) and NAS (blue) glasses. Room-
temperature *F(Q)*-FSDPs of glasses for CAS (**c**) and NAS (**d**) with compositions close

to the *m*-measured ones. The FSDPs of 70 mol% SiO$_2$ glasses are shifted up by 0.5 for
clarity. The average *MRD*-*NBO/T* correlation of two NAS-lines (blue solid and dotted
lines) (**e**) is used to calculate the *MRD* values of *m*-measured NAS glasses. The
changing trends of *m* (**b**) are inversely correlated with the changing trends of *MRD*
(**e**). The higher *m*-values of CAS glasses (**b**) show lower *MRD* values (**e**); the larger *m*-
slope of NAS (**b**) is also supported by the larger negative *MRD*-slope for NAS (**e**).

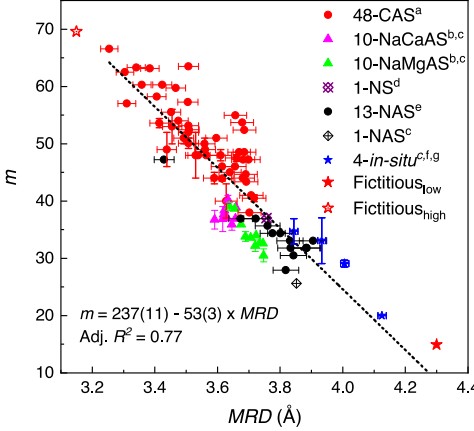

**Fig. 5 | Inverse linear correlation between *m* and *MRD* values for 89 glasses.** The
*MRD* values of 48 CAS glasses (red circles) were calculated using the linear equation
between *MRD* and SiO$_2$ mol% content obtained in Fig. 2a. The *MRD* values of 10 NAS
glasses (black circles) were calculated using the equation, as shown in Fig. 4e. The
*MRD* values of all other glasses were derived from the measured FSDPs of neutron
structure factors. The error bars of calculated *MRD* values were propagated from
errors in the linear equations. The error bars of measured *MRD* values are
approximately the same size as the symbols. The dotted line is a linear fit of the data
for 89 glasses, with $R^2$ = 0.77. [a]: see the resources of Fig. 1; [b]: ref. [29]; [c]: ref. [30]; [d]:
ref. [36], [e]: ref. [24], [f]: ref. [52], [g]: ref. [53].

change in CTE at $T_g$, while fragile glass-forming liquids show a large
change in CTE at $T_g$. This echoes the fact that strong glasses are
expected to be more comparable to their supercooled liquid state than
fragile systems, as they show smaller deviations from equilibrium. The
inverse linear correlation in Fig. 6c, in combination with the positive
correlation in Fig. 6d, reveals the physical origin of the inverse linear
correlation between *m* and RT-*MRD* (Figs. 6e, 5), which is further
interpreted below.

### Physical interpretation of *m*-*MRD* correlations and beyond
In the following, we offer a physical picture to explain the origin of
the inverse linear correlation between the *MRD* value at room tem-
perature and the slope ratio$_{liquid/glass}$ of the *MRD* (and, hence, fragi-
lity). To confirm that the *MRD* can be used as a ring-size index, we
first reveal the valid structure of three CAS glasses by combining
neutron-diffraction experiments and force-enhanced atomic-refine-
ment (FEAR) simulations[28]. Figure 7a presents structure factors *F(Q)*
for three CAS glasses, comparing the neutron-measured *F(Q)* with
simulations by FEAR and MD (for comparison purposes), along with a
similar comparison for the *F(Q)* of fused silica (FS) from ref. [28]. The
fact that the FEAR structure offers an excellent description of the
FSDP of the experimental structure factor of all glasses (significantly
improved as compared to MD), indicates that the FEAR model pro-
vides a realistic description of the glasses' medium-range order. We
then computed the ring-size distribution from the FEAR-simulated
structures using the RINGS code[38] by using Guttman's ring definition.

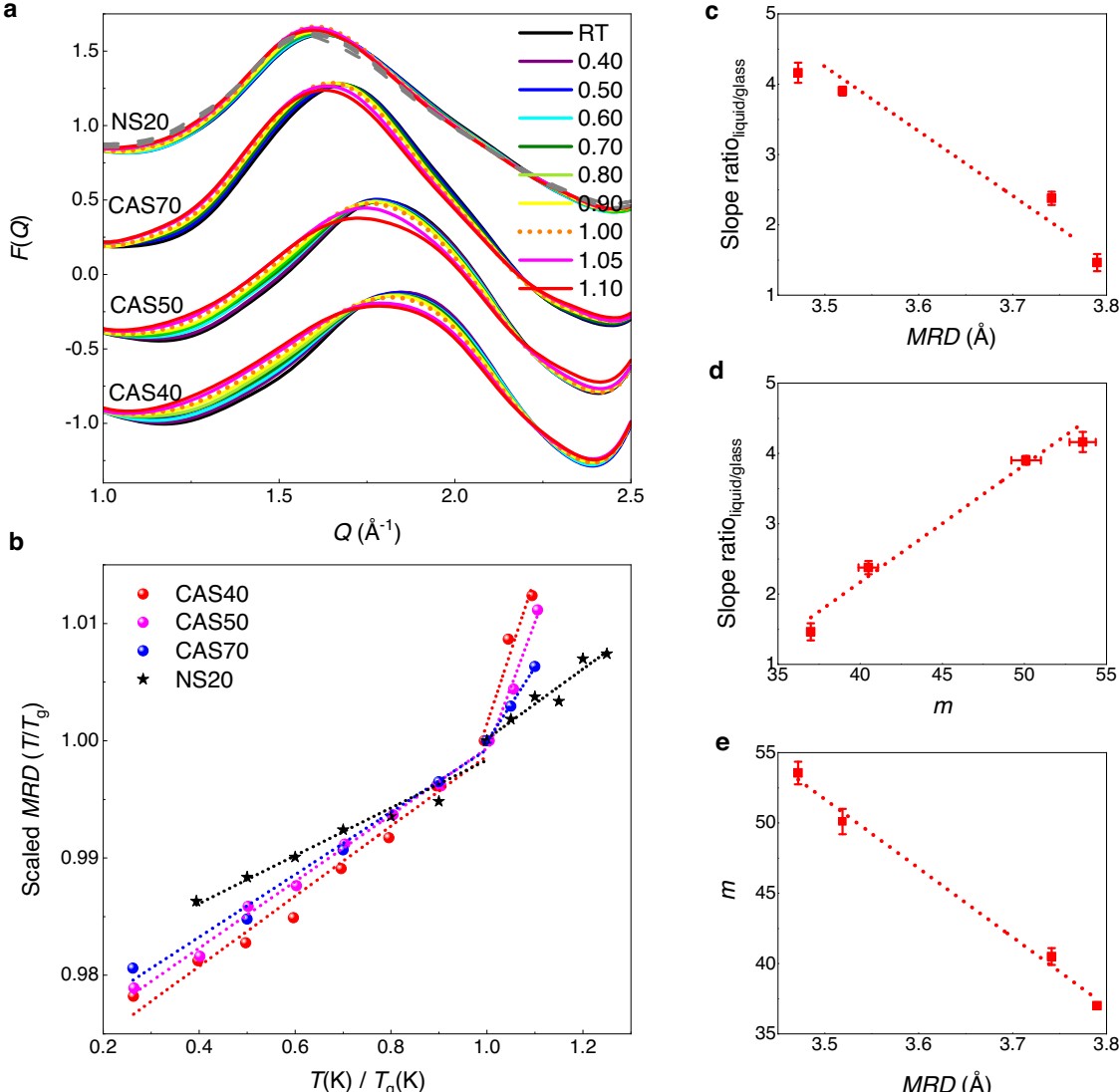

**Fig. 6 | Structural origin of the *m-MRD* correlation revealed by in-situ neutron total-scattering analysis. a** FSDP regions of *F(Q)* of three CAS glasses measured from room temperature up to 1.1 $T_g$ and NS20 glass from RT to 1.25 $T_g$. The RT-*F(Q)* is plotted in black, then the color changes following the rainbow color spectrum, until the *F(Q)* of 1.1 $T_g$ is plotted in red. The three *F(Q)*s with $T$ higher than 1.1 $T_g$ for NS20 glass are plotted by gray dashed lines. Both CAS40 and CAS50 glasses were measured at ten temperatures, as labeled in the figure legend; due to time limitations, CAS70 was only measured at seven temperatures with $T = 0.4$, 0.6 and 0.8 $T_g$ skipped. **b** An analogous OLUA plot of scaled-$MRD_T$/$MRD_{Tg}$ as a function of scaled temperature, $T/T_g$ for the four glasses. The slopes for the *MRD* change in both supercooled-liquid and glass ranges are determined by linear fits, as shown by the dotted lines. Note the significant discontinuous increase of $MRD_T$/$MRD_{Tg}$ on passing through $T_g$. The inverse linear correlation between the through-$T_g$ slope ratio (liquid/glass) of *MRD* as a function of $T/T_g$ and RT-*MRD* (**c**), along with the positive correlation between the through-$T_g$ slope ratio and fragility-*m* (**d**), provide a semi-quantitative justification for the inverse linear correlation between *m* and RT-*MRD* (**e**), as illustrated in Fig. 5 for 89 glasses. The error of the slope ratio is propagated from the linear fitting error in (**b**).

Each glass' ring-size distribution is individually plotted in Fig. 7b as a histogram of absolute ring counts within the simulation. In line with the experimental RingFSDP method, which groups the differently sized rings in three families[27], the grouped ring size distribution (i.e., ≤4-, 5-, and ≥6-membered rings) for all four glasses are compared in one plot (Fig. 7c). The population of small sized, ≤4-membered rings, decreases significantly and systematically with increasing SiO$_2$ content, while, in turn, the population of intermediate 5-membered remains fairly stable and the population of large ≥6-membered rings increase upon increasing SiO$_2$ mol% content. The inverse correlation between *MRD* and the relative percentage of ≤4-membered rings (Fig. 7d) confirms that the *MRD* can be used as a parameter that reflects the percentage of small-sized rings in a glass' structure. As such, low *MRD* values are indicative of the existence of higher amounts of small rings in the glass network. Detailed information

about the FEAR simulations and ring-size distribution calculations is reported in Supplementary Note 4.

The observed inverse linear *m-MRD* correlation clearly demonstrates that differently sized rings exhibit different propensities for deformation. It further indicates that glasses presenting a large fraction of small rings will tend to exhibit a more pronounced variation in their configurational entropy upon increasing temperature, which, in turn, results in higher fragility values. This seems counter-intuitive, since one might expect that the larger size floppy rings should deform more easily than smaller size rigid rings. However, our in-situ neutron-scattering study of fused silica (FS) revealed the opposite, namely, that the intensity of the part of the FSDP associated with small-size rings ($n \leq 4$) decreases more than that associated with large-sized rings with increasing temperature[39]. The experimental observation of smaller size rings being unstable is supported by the following two simulation

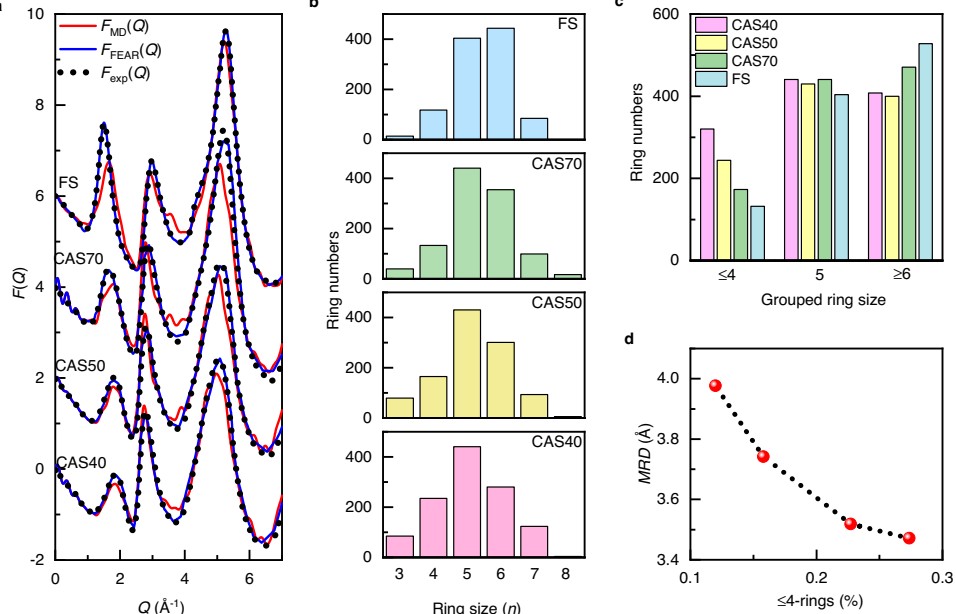

**Fig. 7 | FEAR simulation confirmation that *MRD* can be used to represent the percentage of small-sized rings. a** Measured (black dotted curves) and two simulated, MD (red solid curves) and FEAR (blue solid curves) total reduced structure factors $F(Q)$ for three CAS and FS glasses, with each glass' $F(Q)$s vertically shifted up by 2 for clarity. **b** Guttman's ring-size distribution for each glass. **c** Comparison of grouped ring-size distribution for all glasses. **d** Inverse correlation between *MRD* and relative percentage of ≤4-membered rings.

studies: small-sized rings ($n = 3$ and $4$) are energetically unfavorable, having much higher relative energies as compared to that of 6-membered rings in FS[32]; small-sized rings ($n < 6$) in sodium silicate glasses (e.g. 30 Na$_2$O–70 SiO$_2$) exhibit a significant internal stress on account of their over-constrained topological nature, whereas large-sized rings ($n \geq 6$) do not[33]. Indeed, the mechanical stability of a ring (i.e., floppy/flexible or stressed/over constrained) can be inferred from its number of nodes. For example, a triangle (3-membered ring) is fully determined by the length of the three edges. As such, the angles cannot have arbitrary values and are determined by the lengths of the edges. Hence, 3-membered rings present 3 redundant constraints (e.g., the angles). Similarly, 4- and 5-membered rings present 2 and 1 redundant constraints, respectively. In contrast, with 6-membered rings, all the bonds and angles can have arbitrary values, and hence, such rings (and larger ones) are not over constrained. The fact that the angles in small rings cannot reach their preferred value results in the formation of some internal stress, with the angles being strained so as to enable the ring to close. Such internal stress acts as an elastic-energy penalty that decreases the mechanical stability of small rings.

However, what we present here is an empirical correlation with a qualitative interpretation, not a theoretically derived model to connect the macro-scale fragility with the interatomic interactions determined from *MRD*. Such a model has been theoretically developed for metallic glasses by Krausser, Samwer, and Zaccone (KSZ, hereafter)[40]. This model connects the fragility, viscosity, and high-frequency shear modulus of metallic glasses to the effective ion-ion interaction. Based on the 'shoving' model of the viscosity of glass-forming molecular liquids[41] and the Born-Huang formula for the shear modulus in the harmonic approximation[42], the KSZ model derives an explicit expression for the viscosity as a double-exponential function of $T$ with a power-law exponent $(2 + \lambda)\alpha_T$. The thermal-expansion parameter $\alpha_T$ is determined by the attractive anharmonic part of the effective interaction. For metallic glasses, $\lambda$ has a clear physical meaning, representing the steepness parameter of the repulsive part of the interatomic potential[40]. However, when the KSZ model was applied to fit the viscosity data of oxide network glasses, the power-law exponent $\lambda$ lost its physical meaning and should only be treated as a description

of the potential[43]. Here, we attempt to connect the *MRD* of silicate glasses with the potential parameter, $\lambda$, from the viewpoint of the deformation energy. In ref. [32], the relative energies of different $n$-membered rings are first defined as $\Delta E_n/E_0$, with $E_0$ being an energy scale related to the cohesive energy. The $\Delta E_n/E_0$ values were then calculated by examining the interatomic distances and bond-angle distributions in $n$-membered rings derived from an MD-simulated model of FS. In Fig. 12 of ref. [32], it is shown that both 5- and 6-membered rings have close to a zero energy, while the 4-membered rings have an estimated $\Delta E_4/E_0$ value of 0.06. This shows that the energy of small-sized 4-membered rings is higher than those of 5- and 6-membered rings. Therefore, we connect the relative energy of $n$-membered rings with the potential parameter, $\lambda$; the higher the relative energy of a ring, the higher the steepness of $\lambda$, the higher the fragility. Vice versa, a glass with a higher population of large-sized rings has a lower energy and exhibits a lower fragility, which echoes the finding that "*soft colloids make strong glasses*"[44].

We therefore deduce that glasses comprising higher amounts of unstable small-sized rings (i.e., featuring high penalty energy) tend to deform more with increasing temperature. This can be understood from the fact that, due to their intrinsic mechanical instability, the activation energy associated with the deformation of small rings is smaller than that of large rings since, for instance, such mechanical instability facilitates the buckling of small rings. As such, as the temperature increases, the deformation of small rings is thermodynamically favored at the expense of that of their larger counterparts. This indicates that glasses exhibiting a large fraction of unstable small rings will tend to exhibit a more pronounced variation in their configurational entropy with increasing temperature. Therefore, we propose an inverse correlation between *MRD* and the potential parameter, $\lambda$. Smaller *MRD* values indicate a larger proportion of smaller rings with a higher internal stress/energy, which corresponds to a lower activation energy and consequently a higher potential parameter, $\lambda$, which, in turn, results in higher fragility values.

Furthermore, the deformation of rings (i.e., a change in their shape) is a low-activation-energy event as compared to other potential deformations, such as the breaking of a bond. The external energy

provided by heating is not large enough to overcome the high energy barrier necessary to break or switch a bond; instead, the structure changes in a spatially cooperative manner through torsional motion of $SiO_4$ tetrahedra via Si–O–Si flexible rotations. As such, low-energy deformations should be energetically favored in the vicinity of the glass transition temperature. Structural relaxation is also expected to be largely governed by this deformation mechanism.

## Discussion

In this paper, we report the existence of an inverse correlation between the glass-forming liquid fragility index, $m$, and the glass average medium-range distance ($MRD$). Low $MRD$ values are associated with high $m$ values, and vice versa. Low $MRD$ values are also indicative of a larger proportion of smaller rings. These small rings are more unstable than their larger counterparts. As such, when increasing the temperature above $T_g$, a structure exhibiting a greater proportion of smaller rings will deform more easily. This gives rise to a larger change in shear viscosity with temperature, i.e., a higher fragility index. A glass with a higher proportion of larger, more stable rings will not deform as easily as $T$ increases through $T_g$, so that it takes a larger temperature change to get the same change in viscosity, thereby resulting in a smaller fragility index.

The temperature dependence of the viscosity is an intrinsically kinetic property, but the derivative parameter, fragility, also has a thermodynamic origin that involves the configurational entropy. Such a configurational parameter is related to the atomic structure. In that regard, the two existing approaches, namely, temperature-dependent constraint theory (TDCT) and a coarse-grained model (CGM), focus on different length-scales for fragility interpretation. TDCT describes the glass structure at the short-range level by considering both two-body linear bond-stretching and three-body angular bond-bending constraints. In contrast, CGM expands the length-scale to the network connectivity between rigid polytopes. For silicate glasses, this can be calculated as the average number of bridging oxygen atoms per rigid-unit polyhedron. It is equivalent to the numbers of A–O–A linkages (wherein A are the network-forming cationic species), which is proposed, by CGM, as the weakest linkage giving rise to the structural deformation in the vicinity of $T_g$. Not surprisingly, the A–O–A bond-bending angular constraint is also defined by TDCT as the constraint with the lowest activation temperature, which means that it is the major determining constraint for structural deformation around $T_g$. Therefore, both approaches, TDCT and CGM, focus on the same structural origin, i.e., the weak A–O–A linkages. The difference is that TDCT also considers other minor constraint factors, which are associated with higher activation temperatures[10]. Our $m$-$MRD$ correlation is, indeed, also based on the same structural deformation origin, i.e., A–O–A linkages. However, here, by accounting for the fact that the network rings exhibit some heterogeneity and can exhibit a variety of sizes, we go beyond the mean field, i.e., the average number of constraints per atom captured by TDCT, or average connectivity captured by CGM. In addition, when compared to the CGM approach, we further extend the length-scale to describe the atomic structure—wherein the structure is described in terms of rings (i.e., made of a number of rigid polytopes) rather than as a network of interconnected rigid polytopes. Therefore, we advance the description of the structure-fragility relationship by considering how the propensity for A–O–A linkages to deform depends on the size of the ring to which they belong, since the A–O–A linkages residing in small-size rings exhibit a larger propensity for deformation (due to the internal stress present in small rings) than the ones in large-size rings.

In conclusion, this study reveals the existence of an intimate relationship between medium-range distance and fragility in silicate glasses. Finding a structural origin for the rate of the dynamical arrest on cooling to $T_g$ can facilitate the design of new glass formulations with optimized melting, forming, and relaxation behaviors.

## Methods

### Fragility-index measurements and 95 fragility-index data

The conventional way to determine the fragility index is to measure both the low- and high-temperature ranges of the viscosity curve, which typically covers over ten orders of magnitude of viscosity values. Low viscosities ($\eta < 10^5$ Pa s) obtained above the liquidus temperature can be measured by a concentric-cylinder viscometer, whereas high viscosities ($\eta = 10^6$–$10^{12}$ Pa s) can be measured by the deformation rate under constant stress, including beam-bending (BBV) and ball-penetration viscometry (BPV) methods for the viscosity range in the vicinity of $T_g$ ($\eta > 10^{11}$ Pa s for BBV and $\eta = 10^{10}$–$10^{12}$ Pa s for BPV), and parallel-plate viscometry (PPV) in the softening-point range, which is defined at the viscosity value $\eta = 10^{6.6}$ Pa s[21]. Crystallization and volatilization of melts can hinder high-temperature viscosity measurements, while particular sample-size/shape demands and long measurement times tend to impede low-temperature viscosity experiments[3]. The fragility value is then obtained by fitting the full range of the viscosity-temperature curve with an equation that defines the temperature dependence of the equilibrium viscosity. Many equations have been proposed for this purpose, such as the earliest empirical Vogel–Fulcher–Tammann (VFT) equation[45], the theoretically derived Adam-Gibbs (AG) equation[13], the Avramov–Milchev (AM) equation[46], the MYEGA equation[47], and the recent modified elastic model[30]. It is not surprising that different fragility values can be obtained by fitting different equations to the same experimental viscosity curve. The fragility values determined from viscosity measurements, also called the kinetic fragility, are denoted as $m_{vis}$.

Another direct method to determine $m_{vis}$ is to conduct isothermal equilibrium viscosity measurements by either indentation[17] or three-point beam-bending[35] methods around the glass transition. The temperature dependence of the viscosity becomes Arrhenius-like for all glass-forming liquids in a narrow viscosity range near $T_g$. Therefore, the slope of an OLUA · $\log_{10}\eta$ vs. $T_g/T$ plot yields the fragility index as defined in Eq. (1). This method is relatively more challenging due to the proximity to the equilibrium-to-nonequilibrium transition. Therefore, these measurements must ensure that the glass is in the equilibrium condition when the data used for the viscosity calculation are collected. In addition, temperature accuracy and precision play a big role in these measurements, as they are conducted in a narrow temperature range. If all these challenges are overcome, direct and more accurate fragility-index values can be obtained, as shown for the four CAS glasses in Supplementary Note 3.

Alternatively, the fragility index can be estimated by differential-scanning-calorimetry (DSC) analyses utilizing the thermal-history dependence of the calorimetric fictive temperature. This is based on the premise[3] that the activation energy for viscous flow is equivalent to the activation energy for enthalpy relaxation obtained from the heating/cooling rate ($q_c$) dependence of the fictive temperature ($T_f$). Similar to the different fitting equations for viscosity curves, various methods have been proposed to derive the fragility index from DSC measurements, as summarized in ref. 3. All the methods are based on the same inherent assumption that the correlation of $\log(1/q_c)$ ~ $1/T_f$ is Arrhenius-like near $T_g$. An actual non-Arrhenius deviation leads to some discrepancy between the values for calorimetric fragility ($m_{DSC}$) and kinetic fragility ($m_{vis}$). An empirical equation ($m_{vis} = 1.289 m_{DSC} - 4.326$) has been proposed to correct the measured $m_{DSC}$ values to $m_{vis}$ values for silicate glasses[3], even though there is no consensus on the origin of the discrepancy. In addition, our goal is to understand and explain the relationship between the structure and the fragility of glasses, but not to propose a universal empirical model for $MRD$ and $m$. Thus, the small inaccuracies in the $m$ data do not lead to a change in our observations or the conclusion as demonstrated in Supplementary Note 5.

We establish our conclusions, based on a total of 95 fragility-index values: 88 are published data and 7 were measured in this study using

the isothermal equilibrium viscosity method. As listed in Supplementary Tables 1, 3, 5, 6 and 7, they are categorized into four compositional groups, with the types of fragility measurement and fitting methods specified. The kinetic fragility index, $m_{vis}$, was used to obtain a structural-parameter correlation among all the glasses. Fragility values derived from the Avramov (AV) equation were reported as the fragility index - $\alpha$[22,23] which were converted to the kinetic fragility, $m_{vis}$, with the equation specified in Supplementary Table 1. The value of the calorimetric fragility, $m_{DSC}$, reported in 2008[24] was corrected to $m_{vis}$ using the above empirical equation developed in 2017[3].

As listed in Supplementary Table 3, combined viscometry and dilatometry measurements were conducted for all 20 sodium aluminosilicate (NAS) glass-forming liquids, containing either magnesium or calcium (Mg-NAS and Ca-NAS). Their fragility values were first determined by MYEGA fitting[29]. Later, a modified elastic model[30] was proposed and applied to 8 of the 20 viscosity curves; a new set of fragility values were then derived, which are 2 to 5% lower than the MYEGA-fitted values. Such discrepancies are also observed for Corning Jade™ and NIST 710a glasses (Supplementary Table 6), which are either caused by the different fittings to the same full-range viscosity curve, or the different measurement methods used to obtain the full-range curve or the near-$T_g$ linear plot. We discarded the 20 $m_{DSC}$ data reported in ref. [21] due to their very large discrepancy: the $m_{DSC}$ value of the glass-forming liquid with the composition (in mol%) 60.6 $SiO_2$−20.1 $Al_2O_3$−19.3 CaO on the charge-balanced line ($NBO/T = 0$) was reported as being 32, while in the same paper, the glass-forming liquid with a very similar composition on the constant 20 mol% CaO compositional line (60.3 $SiO_2$−19.7 $Al_2O_3$−19.6 with other impurity oxides <0.4 mol%) showed an $m_{DSC}$ value of 54. For the rest of the glass-forming liquids with multiple $m$-values, the averages were calculated, and their standard-deviation values were used as error bars. Without knowing the 'true' fragility values, we cannot evaluate or even discuss which method yields the most accurate result. Instead, we present the fact that experimentally derived fragility values are scattered. This reinforces the importance of searching for the structural origin of fragility, which could then provide a complementary technique for fragility characterization.

## Neutron total-scattering analyses

Time-of-flight (TOF) neutron-scattering measurements were performed on the Nanoscale-Ordered Materials Diffractometer (NOMAD) at the Spallation Neutron Source (SNS), Oak Ridge National Laboratory[48]. Relevant experimental details were reported in ref. [27]. All the structure factors used in this study were normalized to an absolute scale, utilizing the low-$r$ region of $G(r)$ criterion, as described in ref. [49].

We recently developed a method, namely, RingFSDP, to experimentally quantify the ring-size distribution of silicate glasses from the first sharp diffraction peak (FSDP) of the neutron-scattering structure factor[27]. The FSDP can be deconvolved into three modified Gaussian peaks with fixed positions using a Fourier transformation – fitting – back-Fourier transformation method. Each peak is ascribed to a certain family of rings: large rings (≥ 6-membered) centered at low $Q$; medium rings (5-membered) centered at intermediate $Q$; and small rings (≤ 4-membered) centered at large $Q$. The FSDP deconvolution, as well as ring-size distribution calculation, was performed using the Python program RingFSDP.

Since the shape of the FSDP is generally broad and asymmetric, the following two practices should be applied to ensure a reliable FSDP deconvolution. First, the $F_{FSDP}(Q)$ in reciprocal space needs to be Fourier transformed into its real-space representation, $I_{FSDP}(r)$, since the latter expands the signal and allows for more reliable model fitting. Second, the deconvolution of the FSDP is performed through the fitting of $I_{FSDP}(r)$ by three 'compressed' exponentially decaying sine waves in real space, which corresponds to the three modified Gaussian peaks in reciprocal space. This fitting model was developed empirically from the fitting of the $I_{FSDP}(r)$ profiles of 81 aluminosilicate glasses[27]. It was found that, in most fittings, the periodicities of the three sine waves generally converge well to constant values, i.e., 3.15 ± 0.01, 3.70 ± 0.03, and 4.30 ± 0.04 Å, respectively, where the mean and standard-deviation values were calculated from fitting to the data of 81 glasses. Comparing with the ring-structure information of crystalline $SiO_2$ polymorphs, we postulate that these three characteristic periodicities ($r = 3.15$, 3.70, and 4.30 Å) correspond to the typical effective diameters of small (≤ 4), intermediate (= 5), and large rings (≥ 6), respectively[27].

Mathematically, the three periodicities of the sine waves in real space making up $I_{FSDP}(r)$ correspond to the three Gaussian peak positions of the $F_{FSDP}(Q)$ deconvolution in reciprocal space, with $Q$-values of 2.00, 1.70, and 1.46 Å$^{-1}$, respectively. More importantly, these three discernible $Q$-values are confirmed for both Jade and fused-silica structural models generated from force-enhanced atomic-refinement (FEAR) simulations[28]. The positions of the FSDPs of the three grouped structure factors (i.e., ≤4, = 5, and ≥ 6-membered rings) calculated from FEAR-based structures exhibit very good agreement with the three fixed-$Q$ values that were empirically derived from the FSDP deconvolution of 81 silicate glasses[27]. This confirmation not only provides a strong validation of the RingFSDP method, but also implies that a reliable deconvolution of the FSDP can be achieved by involving only six fitting parameters (i.e., the intensities and widths of the three Gaussian distributions) rather than nine (i.e., if three additional fitting parameters were to be needed for the positions).

In the reciprocal-space structure-factor function, $S(Q)$, the integrated area of each peak is proportional to the absolute number of such specified size rings, wherein the shape of the ring exhibits a certain minimum level of ordering (i.e., poorly-ordered rings are assumed not to contribute to the peak).

In this study, room-temperature (RT) neutron total-scattering patterns were collected for 56 silicate glasses, including 27 CAS glasses, one sodium silicate (NS), five sodium aluminosilicate (NAS) glasses, 20 Mg-NAS and Ca-NAS glasses and three industrially-relevant glasses. The compositions, RingFSDP fitting and *MRD* values of CAS and NAS glasses are listed in Supplementary Tables 2, 4, while information on the Mg-NAS and Ca-NAS glasses are listed in Supplementary Table 3, together with their fragility data.

Three charge-balanced CAS glasses and one binary NS glass (see Supplementary Table 7) were studied by in-situ high-temperature neutron total-scattering analyses at temperatures from RT up to 1.25 $T_g$, with a special focus on the near-$T_g$ range. Both CAS40 and CAS50 glasses were measured at ten temperatures, as labeled in the legend of Fig. 6a; due to time limitations, CAS70 was only measured at seven temperatures with $T = 0.4$, 0.6 and 0.8 $T_g$ being skipped. The medium-range structural change was derived from analyses of the FSDP in the structure factor and is listed in Supplementary Table 8. The measurements and data-analysis methods have been published in ref. 31 and will not be repeated here.

## Data availability

All data generated or analyzed during this study are included in this published article (and its supplementary information files). Experimental neutron diffraction structure factors have been deposited on Materials Data Facility titled as Neutron-diffraction structure factors of silicate glasses (https://doi.org/10.18126/2ntz-4tqe[50]).

## Code availability

The Python program RingFSDP is available from the NOMAD beamline of Oak Ridge National Laboratory (ORNL) (https://code.ornl.gov/mth/ts-tools)[51].

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

## Acknowledgements
Y.S. acknowledges the Scientific User Facilities Division at Oak Ridge National Laboratory's Spallation Neutron Source, Office of Basic Energy Sciences, US Department of Energy. M. B. acknowledges funding from the National Science Foundation under Grants No. CMMI-1762292 and DMR-1928538.

## Author contributions
Conceptualization: Y.S.; Methodology: Y.S., J.N., O.G., N.J.S.; Investigation: all the authors; Modeling: Q.Z., M.B.; Visualization: Y.S.; Writing—original draft: Y.S.; Writing—review & editing: Y.S., M.B., S.R.E., O.G., B.D., D.C.A.

## Competing interests
The authors declare no competing interests.
