## [Peer Review File · Nature Communications]

Revealing the Relationship between Liquid Fragility and Medium-Range Order in Silicate GlassesREVIEWER COMMENTS

Reviewer #1 (Remarks to the Author):

The ms by Shi et al discusses a correlation between the liquid fragility introduced by Angell et al and a medium range order (MRD) defined by the authors due to the structure factor of silicate glasses. In the discussion the authors find unstable small rings which seem to be easily deformed upon increasing temperature and produce therefore a high fragility index versus large -more stable- rings depending on the composition etc of these glasses.

The ms presents a very good set of data, carefully prepared samples and a solid discussion of the findings. In fig 5 the authors demonstrate a linear relationship of the fragility index m versus the MRD parameter as discussed in the methods section. The interpretation follows the rule that small MRD have high m and vice versa by discussing the mechanical stability of the small rings as indicated above. The data are limited to network glasses and do not include other materials like alcohols, metallic glasses, polymers etc.

Here starts my concerns: first the authors need to realise that there is an ongoing discussion on the importance of the local to medium size shear modulus on the viscosity and the fragility index getting from the viscosity. J. Dyre, J. Langer, I. Procaccia, W. Johnson among others published quite a number of papers where they showed how the viscosity and its slope near T_g depends on that local to medium range value of the shear modulus. S. Alexander wrote a large review on it and newer publications even produced a complete microscopic model for the viscosity starting from the Born-Huang approach for the crystal-liquid transition to give an analytical equation for the viscosity and the fragility of liquids and glasses (A. Zacccone et al. PNAS 2015). Probing this so-called KSZ model on more than 30 very different glassforming systems showed a nearly perfect fit to the experimental data and demonstrate a linear fragility versus steepness parameter λ (P. Lunkenheimer et al JCP 2020).

Now the most interesting question arises how this steepness parameter correlates to the MRD proposed by the authors??

Is m just depending on the SiO_2 content (Fig 2a) and its density fluctuations as recently discussed by Novikov (PRE 2022)? If one includes the AlO_2 glasses the scatter of the data increases and a rather complex behaviour as discussed carefully by the authors occurs. In general the ms is a very solid presentation of good data in combination with a rather old and limited model. More verbal than hard analytical equation based discussion is presented and some aspects might be even wrong. Nevertheless the data corroborates the fact presented by D Weitz some years ago that "soft colloids make strong glasses". Therefore I suggest a major revision of the ms but hope, these very good data gets published for the broader community.

Reviewer #2 (Remarks to the Author):

The structural origin of fragility (m) is an important topic in the field of glassy physics. It is very helpful if the new results of this topic has been convinced. The authors reported an inverse correlation between m and medium-range distance (MRD). It is very interesting to demonstrate such a relationship. However, there are a few concerning points as listed in the following should be addressed.

there are should be the specific reasons are as follows:

(1) The authors attempt to explain the structural origin of kinetic fragility from the viewpoint that how the propensity for the weak A-O-A linkages to be deformed in rings of different sizes. According to the author's interpretation, the existence of strong internal stress in the small-sized rings leads to the increase of the number density of the weak A-O-A linkages, thus the system's ability to resist deformation becomes weak, resulting in a faster change rate of viscosity with temperature variation (i.e., the higher fragility value). The author's view is contrary to the conventional understanding of the relationship between fragility and deformability. Novikov and Sokolov (Nature 431, 961 2004) found a positive linear relationship between the value of m and the ratio of the instantaneous bulk and shear moduli (K_{∞}/G_{∞}) in some oxide and organic glasses. Traditionally, it has been recognized that the small-sized rings can effectively improve the packing density of the glass system, and thus have higher shear deformation resistance. The viscosity of the supercooled liquid of the dense-packing system changes slowly with temperature variation, showing a strong liquid behavior with a low m value. The authors held the opposite view, but there is no detailed experimental evidence or theory deducing to prove it. In addition, the authors said that the deformation of rings (i.e., a change in their shape) is a low-temperature energy event through the flexible rotation of Si-O-Si. This is a localized deformation mode, which is manifested in the form of structural relaxation in the glass system. However, the fragility describes the slope ratio of viscosity changes with temperature when all supercooled liquid atoms collectively move. The size of deformation unit (weak A-O-A linkages) is much smaller than that of collective movement. Thus, I have serious doubts about the explanation on the structural origin of fragility from the perspective of very locally deformation mode.

(2) I have some concerns about the accuracy of m obtained by DSC measurement. Fragility m is defined as an index to characterize how fast the viscosity of the glass-forming liquid increases while temperature approaches T_g , it should be directly determined by viscosity measurements. DSC method is easy to bring great error to the fragility value. For example, Yue et al. (JNCS 456, 95 2007) established an empirical formula between calorimetric (mDSC) and kinetic (mvis) fragility, which is only valid for a certain system of glasses. In this paper, the empirical formula is used to convert mDSC into mvis, which can easily bring uncertainties of fragility data. By the way, the m value of CAS40 in Table S7 is crucial to the validity of discussion section, but it is not known how it was obtained.

(3) I found that the fragility data of CAS50 in Fig. 6 is wrong, this has a serious impact on the authenticity of the inverse correlation between m and MRD. The authors use in-situ neutron scattering experiment to approve the inverse correlation between m and MRD. The authors found a positive linear correlation between m and slope ratio (liquid/glass) in Fig. 6d, and an inverse linear correlation between m and MRD in Fig. 6e for CAS40, CAS50, and CAS70. However, I found that the experimental data were not able to support the main idea of this paper, because the authors had made a serious mistake in the fragility m data. Specifically, m of CAS50 is directly given as 53.9 in Table S7, while m of CAS50 is 43.9 according to the fragility data in Table S1. After the correction of m , it is easy to see that there is no good linear relationship between m and slope ratio (liquid/glass), and the linear correlation between m and the measured MRD is also very poor. Therefore, the inverse correlation between m and MRD is not well verified by the results of neutron scattering experiments.

(4) The m value of the Fictitioushigh glass-forming liquid is assumed to be 70. But as far as I know, there are some oxide glasses that have m values much higher than 70. Thus, I think the assumption of m value of the Fictitioushigh glass-forming liquid is not reasonable.

(5) As for the MRD data of NAS glasses, they were obtained based on the assumption that

the linear correlation between MRD and NBO/T is valid in the narrow composition range. However, there are only three data points in Fig. 4e, and the linear correlation between MRD and NBO/T is very poor. Therefore, I think that the MRD data of NAS glasses may have great inaccuracy.

(6) What's the physical meaning of slope ratio (liquid/glass) of MRD? Fragility is defined as the slope ratio of viscosity when temperature is approaching T_g . The slope ratio of MRD between liquid and glass is a dimensionless value, which cannot be related to fragility. Why did not the authors directly compare the slope of MRD around T_g with fragility?

(7) Finally, there are some minor mistakes. Abbreviations of words like CAS, NAS, and CTE do not give the full names. There is a double "[3]" error on line 465.

Based on the above considerations, I believe it is not suitable to consider to publish the manuscript in the NC journal.

Reviewer #3 (Remarks to the Author):

This paper reports the relationship between the fragility (m) and the medium-range order in silicate glasses. The major experimental finding is the inverse linear relationship between the fragility and the average medium-range distance MRD deduced from the first sharp diffraction peak. This was based on ~80 data points from CAS, NAS, and other silicate glasses. The authors went on to argue that the MRD value is related to the excess of small network rings, which they claim are less stable, and later on discussed the results in the context of coarse-grained model (aka the linkage of A-O-A networks).

The paper is well-written and a pleasure to read. The experimental results are solid. However, to me, the paper fell short of providing a convincing explanation for the observed correlation. It would be more effective if the authors have done some simulations, as they have in their earlier publications, to elucidate the origin of the m -vs-MRD correlation. The presented experimental results with heating are nice but they do not give definitive answers. After all, there are only three data points (Fig. 6). Simulations with a few model silicate systems, especially numerical results showing how low MRD values are correlated with the excess amount of small rings, would significantly strengthen the manuscript.

Some other comments.

1. As MRD is central to the discussion, it would be helpful to the readers to outline in the main text how MRD is determined (Eq. (2-3) in the Methods) and how it is treated as a length-weighted ring-size index.
2. Is there an estimate of the energy barrier for deforming the small rings (say at room temperature)?

First of all, we would like to thank all the reviewers for their careful reviews and helpful comments. We believe that the comments not only have helped us to improve/clarify our manuscript, but also inspired us to think deeper and go beyond the experimental method presented in the initial manuscript.

Below are our point-by-point replies to the specific comments raised by the reviewers, including details on how we have revised the manuscript.

For clarification, the format is defined below:

Reviewers' comments (gray highlighted)

Authors' replies (blue font)

Changes made in the manuscript (black Arial font with double line spacing)

Reply to Reviewer #1's comments

Reviewer #1: The ms by Shi et al discusses a correlation between the liquid fragility introduced by Angell et al and a medium range order (MRD) defined by the authors due to the structure factor of silicate glasses. In the discussion the authors find unstable small rings which seem to be easily deformed upon increasing temperature and produce therefore a high fragility index versus large -more stable- rings depending on the composition etc of these glasses.

The ms presents a very good set of data, carefully prepared samples and a solid discussion of the findings. In fig 5 the authors demonstrate a linear relationship of the fragility index m versus the MRD parameter as discussed in the methods section. The interpretation follows the rule that small MRD have high m and vice versa by discussing the mechanical stability of the small rings as indicated above. The data are limited to network glasses and do not include other materials like alcohols, metallic glasses, polymers etc.

Authors' reply: We thank the Reviewer #1 for the clear summary of our manuscript. It is correct that our study only applies to silicate glasses. We are looking for an atomic-scale fingerprint that governs the rate of dynamic arrest at the glass transition. We do not intend to propose a universal structural descriptor for fragility, because we believe that glasses with very different structures (e.g., non-network glasses) should exhibit different structure-dynamics mechanisms. Moreover, we doubt the existence of a universal rule that is generically applicable for all structural types of glassy materials.

Changes made in the manuscript: No change is made in the manuscript.

1. Here starts my concerns: first the authors need to realize that there is an ongoing discussion on the importance of the local to medium size shear modulus on the viscosity and the fragility index getting from the viscosity. **J. Dyre, J. Langer, I. Procaccia, WI Johnson** among others published quite a number of papers where they showed how the viscosity and its slope near T_g depends on that local to medium range value of the shear modulus. **S. Alexander** wrote a large review on it and newer publications even produced a complete microscopic model for the viscosity starting from the Born-Huang approach for the crystal- liquid transition to give an analytical equation for the viscosity and the fragility of liquids and glasses(**A.Zaccone et al. PNAS 2015**). Probing this so called KSZ model on more than 30 very different

glass forming systems showed a nearly perfect fit to the experimental data and demonstrate a linear fragility versus steepness parameter λ (P.lunkenheimer et al JCP2020). Now the most interesting question arises how this steepness parameter correlates to the MRD proposed by the authors??

Authors' reply: We highly appreciate Reviewer #1's inspiring comments, which led us into the field of theoretical condensed matter to connect the atomic interaction derived from structure to shear modulus, then to the viscosity. Although we are not able to derive an explicit model for silicate glasses based on their ring structure, like the KSZ model for metallic glasses, we do attempt to make a connection between our MRD fingerprint and the potential parameter λ from the viewpoint of deformation energy.

Changes made in the manuscript: we add two paragraphs in the section of "*Physical interpretation of m-MRD correlations and beyond*"(P21-23).

However, what we present here is an empirical correlation with a qualitative interpretation, not a theoretically derived model to connect the macro-scale fragility with the interatomic interactions determined from MRD. Such a model has been theoretically developed for metallic glasses by Krausser, Samwer and Zaccone (KSZ, hereafter) [42]. This model connects the fragility, viscosity, and high-frequency shear modulus of metallic glasses to the effective ion-ion interaction. Based on the 'shoving' model of the viscosity of glass-forming molecular liquids [43] and the Born-Huang formula for the shear modulus in the harmonic approximation [44], the KSZ model derives an explicit expression for the viscosity as a double-exponential function of T with a power-law exponent $(2 + \lambda)\alpha_T$. The thermal-expansion parameter α_T is determined by the attractive anharmonic part of the effective interaction. For metallic glasses, λ has a clear physical meaning, representing the steepness parameter of the repulsive part of the inter-atomic potential [42]. However, when the KSZ model was applied to fit the viscosity data of oxide network glasses, the power-law exponent λ lost its physical meaning and should only be treated as a description of the potential [45]. Here, we attempt to connect the MRD of silicate glasses with the potential parameter, λ , from the viewpoint of the deformation energy. In Ref. [32], the relative energies of different n -membered rings are first defined as $\Delta E_n/E_0$, with E_0 being an energy scale related to the cohesive energy, which is derived as 1744 kJ/mole for FS [46], corresponding to 18 eV per SiO_2 molecule. The $\Delta E_n/E_0$ values were then calculated by examining the interatomic distances

and bond-angle distributions in n -membered rings derived from an MD-simulated model of FS. In Fig.12 of Ref. [32], it is shown that both 5- and 6-membered rings have close to a zero energy, while the 4-membered rings have an estimated $\Delta E_4/E_0$ value of 0.06, which corresponds to a penalty energy of 4.3 eV for a 4-membered ring composed of 4 SiO₂ molecules. This shows that the energy of small-sized 4-membered rings is 4.3 eV higher than those of 5- and 6-membered rings. Therefore, we connect the relative energy of n -membered rings with the potential parameter, λ ; the higher the relative energy of a ring, the higher the steepness of λ , the higher the fragility. Vice versa, a glass with a higher population of large-sized rings has a lower energy and exhibits a lower fragility, which echoes the finding that “*soft colloids make strong glasses*” [47].

We therefore deduce that glasses comprising higher amounts of unstable small-sized rings (i.e., featuring high penalty energy) tend to deform more with increasing temperature. This can be understood from the fact that, due to their intrinsic mechanical instability, the activation energy associated with the deformation of small rings is smaller than that of large rings since, for instance, such mechanical instability facilitates the buckling of small rings. As such, as the temperature increases, the deformation of small rings is thermodynamically favored at the expense of that of their larger counterparts. This indicates that glasses exhibiting a large fraction of unstable small rings will tend to exhibit a more pronounced variation in their configurational entropy with increasing temperature. Therefore, we propose an inverse correlation between MRD and the potential parameter, λ . Smaller MRD values indicate a larger proportion of smaller rings with a higher internal stress/energy, which corresponds to a lower activation energy and consequently a higher potential parameter, λ , which, in turn, results in higher fragility values.

2. Is m just depending on the SiO₂ content (Fig 2a) and its density fluctuations as recently discussed by **Novikov (PRE 2022)**? If one includes the AlO₂ glasses the scatter of the data increases and a rather complex behaviour as discussed carefully by the authors occurs.

Authors' reply: We echo Reviewer #1's concern that m may just simply correlate with the SiO_2 mol%, as we initially observed for CAS glasses. We have demonstrated that the inverse correlation between m and SiO_2 mol% for the 48 CAS glasses is due to the linear correlation between MRD and SiO_2 mol% for the CAS glass system. Such an MRD - SiO_2 mol% linear correlation is only applicable for the CAS glass system. Our neutron-scattering data show that no such correlation exists for the wide range of the sodium aluminosilicate (NAS) glass system.

To further illustrate the point, we plot both the m - MRD and m - SiO_2 mol% correlations for all 88 glasses (as shown below). An inverse correlation between m and SiO_2 mol% is also found for all 88 glasses, with a fair coefficient of determination $R^2 = 0.60$. However, we argue that this apparent correlation mainly originates from the 48 CAS glasses. We then remove the 48 CAS glasses and only plot the m - MRD and m - SiO_2 mol% correlations for the remaining 40 glasses. An inverse correlation of m with MRD still holds valid, with a slightly lower R^2 value of 0.61. In contrast, no correlation of m and SiO_2 mol% is observed for the remaining 40 glasses. This confirms that m does not depend on SiO_2 mol% for silicate glasses, except for the CAS system.

Change 1 made in the manuscript: a sentence is added in the section "*Universal m - MRD correlation for 88 glasses*" (P14).

Main text

In section 2 of the Supplementary Document, we further confirm that such inverse linear correlation only exists between m - MRD , not between m - SiO_2 mol%.

Change 2 made in the manuscript: Section 2 has been added in the Supplementary Document.

Supplementary Document

2. No inverse correlation between m - SiO_2 mol% for the non-CAS glasses

To further confirm that m depends on MRD but not on SiO_2 mol%, we plot both the m - MRD and m - SiO_2 mol% correlations in **Error! Reference source not found.** below. In panel (a), an inverse correlation between m and MRD with $R^2=0.77$ is observed for all 88 glasses. In panel (b), an inverse correlation between m and SiO_2 mol% is also found for all 88 glasses, with a fair coefficient of determination $R^2 = 0.60$. However, we argue that this apparent correlation mainly originates from the 48 CAS glasses. We then remove the 48 CAS glasses and only plot the m - MRD and m - SiO_2 mol% correlations for the remaining 40 glasses in panels (c) and (d). In panel (c), an inverse correlation of m with MRD still holds valid, with a slightly lower R^2 value of 0.61.

In contrast, no correlation of m and SiO_2 mol% is observed for the remaining 40 glasses in panel (d). This confirms that m does not depend on SiO_2 mol% for silicate glasses, except for the CAS system.

Fig. 1 | Comparison of the m -MRD and m - SiO_2 mol% correlations. (a) Inverse correlation between m and MRD with $R^2 = 0.77$ for all the 88 glasses. (b) Inverse correlation between m and SiO_2 mol% with $R^2 = 0.60$ for all the 88 glasses. (c) Inverse correlation between m and MRD with $R^2 = 0.61$ for 40 glasses, with the 48 CAS glasses being removed. (d) No correlation between m and SiO_2 mol% with $R^2 = 0.25$ for all 40 glasses, with the 48 CAS glasses being removed.

The Novikov (PRE 2022) paper referred by Reviewer #1 correlates the *shear-modulus* fluctuation with fragility; the density fluctuations on the nanometer scale were actually neglected in the P3 of their paper (the paragraph above Equation (18)). We are not able to comment in detail on this paper, but we don't think that our work is related to this paper.

Changes made in the manuscript: No change is made in the manuscript.

3. In general the ms is a very solid presentation of good data in combination with a rather old and limited model. More verbal than hard analytical equation based discussion is presented and some aspects might be even wrong. Nevertheless the data corroborates the fact presented by D Weitz some years ago that "soft colloids make strong glasses".

Authors' reply: We agree with Reviewer #1's comment that our study is an empirical observation based on a large set of experimental data; the interpretation is very qualitative, even though we do attempt to correlate with the physics model introduced by the Reviewer #1. We hope that our findings can stimulate the interest of other physicists to develop a quantitative analytical model.

We correlate the lower-energy large-sized rings with soft colloids and cite D Weitz's Nature paper in our manuscript.

Changes made in the manuscript: a sentence is added in P22.

Vice versa, a glass with a higher population of large-sized rings has a lower energy and exhibits a lower fragility, which echoes the finding that "*soft colloids make strong glasses*" [47].

[47] J. Mattsson, H.M. Wyss, A. Fernandez-Nieves, K. Miyazaki, Z. Hu, D.R. Reichman, D.A. Weitz, "Soft colloids make strong glasses," *Nature*, vol. 462, pp. 83-86, 2009.

Therefore I suggest a mayor revision of the ms but hope, these very good data gets published for the broader community.

Authors' reply: We thank Reviewer #1 for his/her encouraging words and hope to share our findings with the broader community to derive a physics-based model that can describe the ring-shape deformation at the vicinity of the glass transition.

Reply to Reviewer #2's comments

Reviewer #2: The structural origin of fragility (m) is an important topic in the field of glassy physics. It is very helpful if the new results of this topic has been convinced. The authors reported an inverse correlation between m and medium-range distance (MRD). It is very interesting to demonstrate such a relationship. However, there are a few concerning points as listed in the following should be addressed.

For clarity, we break Reviewer #2's first comment into four parts and reply point by point.

1. **A-1:** The authors attempt to explain the structural origin of kinetic fragility from the viewpoint that how the propensity for the weak A-O-A linkages to be deformed in rings of different sizes. According to the author's interpretation, the existence of strong internal stress in the small-sized rings leads to the increase of the number density of the weak A-O-A linkages, thus the system's ability to resist deformation becomes weak, resulting in a faster change rate of viscosity with temperature variation (i.e., the higher fragility value).

Authors' reply: We thank the Reviewer #2 for his/her understanding and clear summary of our manuscript.

A-2: The author's view is contrary to the conventional understanding of the relationship between fragility and deformability. Novikov and Sokolov (**Nature 431, 961 2004**) found a positive linear relationship between the value of m and the ratio of the instantaneous bulk and shear moduli (K_{∞}/G_{∞}) in some oxide and organic glasses.

Authors' reply: Novikov and Sokolov, in their 2004 Nature paper, reported a positive linear relationship between m and the ratio between bulk and shear moduli (K_{∞}/G_{∞}); the latter is linearly related to the Poisson's ratio. This finding has since then been proved invalid in the "Brief Communications Arising" of 2006 Nature (Nature 442, E7-E8 doi: 10.1038/nature04967 (2006)) "because of the unjustifiable preference for an empirical variation of m with elastic properties, and because of the selected use of glasses". When more glasses are considered in the same way, m does not seem to be linearly related to K_{∞}/G_{∞} . It is worth pointing out that Novikov, in his own 2022 PRE paper (PRE 106, 024611 2022), stated that "Note that the last correlation (m and K_{∞}/G_{∞}) does not hold in complex glasses like silicates, borates, etc., in highly fragile polymers, and looks different in bulk metallic glasses." In turn, our results are limited to silicate glasses. Therefore, we will not connect this paper with our work.

A-3: Traditionally, it has been recognized that the small-sized rings can effectively improve the packing density of the glass system, and thus have higher shear deformation resistance. The viscosity of the supercooled liquid of the dense-packing system changes slowly with temperature variation, showing a strong liquid behavior with a low m value. The authors held the opposite view, but there is no detailed experimental evidence or theory deducing to prove it.

Authors' reply: We agree that small-sized rings correlate with a higher packing fraction, which, in turn, leads to a higher modulus. However, the higher modulus is not necessarily correlated with a lower fragility. Although viscosity is defined by the shear movement, which is in turn controlled by the shear modulus, the fragility m is defined as the changing rate, i.e. the slope of viscosity as a function of T at

T_g . Therefore, m should be determined by the slope of the shear modulus as a function of T , instead of the absolute value of the shear modulus.

We disagree with Reviewer #2's statement that "viscosity of ...the dense-packing system...showing a strong liquid behavior with a low m value". We use fused silica (FS) as evidence to prove that this statement is not correct. FS is the **most loosely-packed** silicate glass and contains the largest proportion of large-sized rings, but, in turn, it is also the strongest glass with the lowest m -value. Our view is that the small-sized rings are unstable due to their higher internal stress and energy, which leads to a higher deformation rate and a larger m . Our manuscript is based on large amounts of experimental evidence supporting this view.

A-4: In addition, the authors said that the deformation of rings (i.e., a change in their shape) is a low-temperature energy event through the flexible rotation of Si-O-Si. This is a localized deformation mode, which is manifested in the form of structural relaxation in the glass system. However, the fragility describes the slope ratio of viscosity changes with temperature when all supercooled liquid atoms collectively move. The size of deformation unit (weak A-O-A linkages) is much smaller than that of collective movement. Thus, I have serious doubts about the explanation on the structural origin of fragility from the perspective of very locally deformation mode.

Authors' reply: We agree with Reviewer #2's view that individual A-O-A deformations are local-scale changes, while the viscosity change at T_g involves macro-scale deformations that are the collective movements of essentially all the atoms. However, our local A-O-A deformation mechanism is not in contradiction with the macro-scale collective movement. Instead, it is the local A-O-A deformations that enable and initiate to the macro-collective movements. We have illustrated this view in two separate places of the original manuscript: 1) Introduction of a "coarse-grained" theory: This is based on the idea that, just above T_g , it is the weak A-O-A linkages between rigid units that **first deform to enable viscous flow**, while the bonds and bond angles within the rigid units remain largely undeformed; 2) At the end of "*Physical interpretation of m -MRD correlations and beyond*" section: structure changes in a **spatially cooperative manner through torsional motion of SiO₄ tetrahedra via Si-O-Si flexible rotations**. As such, low-energy deformations should be energetically favored in the vicinity of the glass-transition temperature.

Changes made in the manuscript: Based on the above discussions, no change is made in the manuscript.

2. I have some concerns about the accuracy of m obtained by DSC measurement. Fragility m is defined as an index to characterize how fast the viscosity of the glass-forming liquid increases while temperature approaches T_g , it should be directly determined by viscosity measurements. DSC method is easy to bring great error to the fragility value. For example, Yue et al. (JNCS 456, 95 2007) established an empirical formula between calorimetric (mDSC) and kinetic (mvis) fragility, which is only valid for a certain system of glasses. In this paper, the empirical formula is used to convert mDSC into mvis, which can easily bring uncertainties of fragility data.

Authors' reply: We agree with the reviewer that the ideal method of fragility-index determination should be based on viscosity measurements, as Angell's original definition is based on viscosity. In addition, it can

be argued that the assumption made in calorimetric determinations of the fragility index, which requires the equivalence of the activation energy for enthalpy relaxation and activation energy for viscous flow, is not valid for most glasses. We share the same concerns with the reviewer, even though various studies show relatively close fragility-index values based on the two methods. Therefore, all the fragility-index measurements performed in this study are based on equilibrium viscosity measurements in the vicinity of Angell's definition of T_g which is the temperature at which the equilibrium viscosity is equal to 10^{12} Pa.s. However, equilibrium viscosity measurements in the T_g range are very challenging and tedious, so the studies in the literature consist of various methods. The most common method used in the literature are the calorimetric rate-dependent measurements, equilibrium viscosity measurements around T_g and viscosity three-parameter model fits of a wide range viscosities. Frequency-dependent methods and dilatometric rate-dependent methods have also been used by various research labs as well.

Since it is not possible to obtain and measure every single glass in this study, we also used data from the literature. Unfortunately, some of them had to be calorimetric measurements due to the limited viscosity data in the literature. Considering the possible accuracy problems in the calorimetric data, we employed the empirical relationship of Zheng et al [JNCS 456, 95 2017] obtained from the study of various oxide glass systems, including borates, aluminosilicates, and vanadium tellurites. Even though Zheng et al. did not prove that it was a universal empirical relationship, it can be used as a method to improve the accuracy of calorimetric fragility-index data in silicates considering the diversity of oxide glass systems used in that empirical study.

In addition, our goal is to understand and explain the relationship between the structure and the fragility of glasses, but we are not trying to propose a universal empirical model for MRD and m . Thus, the small inaccuracies in the m data do not lead to a change in our observations or the conclusion. In order to demonstrate the effect of the inaccuracy of fragility-index data based on calorimetric measurements, we plot the same dataset without using the correction used in the plot in the manuscript. The comparison of the two plots clearly shows that the general trend and correlation between the two parameters do not change when we use the calorimetric data as reported in the cited references without employing the correction. The linear fit quality is slightly worse than the original case, but the difference between the two fits can be considered negligible.

Also, we would like to point out that similar correlations between fragility index and dynamic or structure properties have always used literature data based on various methods. For instance, a paper co-authored by Austen Angell, who introduced the concept of fragility and the fragility index to the glass field, used six different methods to obtain the fragility index [Bohmer et al. J. Chem. Phys. 99, 4201 (1993)]. It is one of the most highly cited paper in his career (2553 citations as of 2022) and, similar to our case, their goal was to show a general correlation between the two parameters rather than introducing an empirical relationship. In conclusion, we can say that the use of calorimetric fragility-index data does not change our findings or conclusions, even though it adds some extra scatter in the global trend.

Change 1 made in the manuscript: a sentence is added in the “**Methods**” section’s “Fragility-index measurements and 95 fragility-index data” (P26-27).

Main text

In addition, our goal is to understand and explain the relationship between the structure and the fragility of glasses, but not to propose a universal empirical model for MRD and m . Thus, the small

inaccuracies in the m data do not lead to a change in our observations or the conclusion as demonstrated in section 5 of the Supplementary Document.

Change 2 made in the manuscript: Section 5 has been added in the Supplementary Document.

Supplementary Document

5. Inaccuracy of DSC-measured fragility data does not affect m - MRD correlation

Since it is not possible to obtain and measure every single glass in this study, we also used data from the literature. Unfortunately, some of them had to be calorimetric measurements due to the limited viscosity data in the literature. Considering the possible accuracy problems in the calorimetric data, we employed the empirical relationship of Zheng et al [9] obtained from the study of various oxide glass systems, including borates, aluminosilicates, and vanadium tellurites. Even though Zheng et al. did not prove that it was a universal empirical relationship, it can be used as a method to improve the accuracy of calorimetric fragility-index data in silicates considering the diversity of oxide glass systems used in that empirical study.

In addition, our goal is to understand and explain the relationship between the structure and the fragility of glasses, but we are not trying to propose a universal empirical model for MRD and m . Thus, the small inaccuracies in the m data do not lead to a change in our observations or the conclusion. In order to demonstrate the effect of the inaccuracy of fragility-index data based on calorimetric measurements, we plot the same dataset without using the correction used in the plot in the manuscript. The comparison of the two plots (Fig. 2) clearly shows that the general trend and correlation between the two parameters do not change when we use the calorimetric data as reported in the cited references without employing the correction. The linear fit quality is slightly worse than the original case, but the difference between the two fits can be considered negligible.

Fig. 2 | Universal inverse linear correlation between m and MRD values for 88 glasses. The fragility-index values of 9 CAS and 13 NAS glasses from DSC measurements were corrected by Zheng's equation (a) or as reported in the original paper without correction (b).

By the way, the m value of CAS40 in Table S7 is crucial to the validity of discussion section, but it is not known how it was obtained.

Authors' reply: We agree that the fragility values of three CAS glasses measured by *in-situ* neutron scattering are crucial to the validity of the discussion, so we made a secondary measurement for the CAS40 and CAS50 glasses, with separate sample preparation, and present all the results in section 3 of the Supplementary Document.

Changes made in the manuscript: please see the changes made in the reply to comment (3) below.

3. I found that the fragility data of CAS50 in Fig. 6 is wrong, this has a serious impact on the authenticity of the inverse correlation between m and MRD. The authors use *in-situ* neutron scattering experiment to approve the inverse correlation between m and MRD. The authors found a positive linear correlation between m and slope ratio (liquid/glass) in Fig. 6d, and an inverse linear correlation between m and MRD in Fig. 6e for CAS40, CAS50, and CAS70. However, I found that the experimental data were not able to support the main idea of this paper, because the authors had made a serious mistake in the fragility m data. Specifically, m of CAS50 is directly given as 53.9 in Table S7, while m of CAS50 is 43.9 according to the fragility data in Table S1. After the correction of m , it is easy to see that there is no good linear relationship between m and slope ratio (liquid/glass), and the linear correlation between m and the measured MRD is also very poor. Therefore, the inverse correlation between m and MRD is not well verified by the results of neutron scattering experiments.

Authors' reply: We sincerely appreciate Reviewer II's careful check and apologize for the serious mistakes that we made in Tables S1 & S7, Fig.6. In this study, we melted four charge-balanced CAS glasses with nominal SiO₂ contents of 40, 50, 60, and 70 mol%. For our first fragility measurement, glass CAS40 was

misabeled as CAS50, and vice versa for CAS40. We then did an ICP analysis and realized the mistake; we made the correct labeling for our *in-situ* neutron measurements. However, we forgot to correct the Tables S1 & S7. To double confirm the fragility data, we re-prepared the correctly labeled glasses CAS40 and CAS50 and measured their fragilities for a second time. The secondary measurement of CAS40 and CAS50 glasses show good reproducibility with the first measurements. We present all the fragility measurement results in section 3 of the Supplementary Document.

Change 1 made in the manuscript: Supplementary Data Tables S1 & S7 have been updated with the correct fragility-index values for CAS40 and CAS50.

Table S1. Composition, *NBO/T* ratio, measurement-fitting equation and value of fragility, and calculated *MRD* of 48 CAS glasses

Series	Composition (mol%)			NBO/T	Fragility				Calculated MRD * (Å)
	SiO ₂	Al ₂ O ₃	CaO		Measurement method	Reported	Conversion or correction method	m -value	
NBO/T=0 this study	40.32	29.82	29.86	0.00	Isothermal equilibrium viscosity measurement near T_g	53.6	Linear fit of Angell plot, no correction	53.6	3.43
	49.61	25.13	25.27	0.00		50.1		50.1	3.50
	60.3	19.9	19.79	0.00		43.9		43.9	3.61
	70.48	14.82	14.7	0.00		40.5		40.5	3.71

Table S7. Composition, *T_g*, fragility, *RT-MRD*, the slope of *in-situ MRD* with *T* change for slope_g, slope_l, and their ratio (slope_l/slope_g). The error in the slope for the *S(Q)*-FSDP area and *MRD* change is determined from linear fitting. The errors for the liquid/glass ratio for *MRD* slopes are calculated by error propagation

Glass I.D.	Composition (mol%)			T_g (K)	Fragility*	RT-MRD (Å)	Slope of in-situ MRD with T		
	CaO or Na ₂ O	Al ₂ O ₃	SiO ₂				Slope _{glass}	Slope _{liquid}	Slope Ratio (liquid/glass)
CAS40	29.86	29.82	40.32	1126.5	53.6(8)	3.47	0.029(2)	0.124(3)	4.2(1)
CAS50	25.27	25.13	49.61	1132.8	50.1(9)	3.52	0.028(1)	0.111(1)	3.90(6)
CAS70	14.70	14.82	70.48	1148.1	40.5(6)	3.74	0.027(2)	0.063(3)	2.38(9)
NS20	20.50	0.00	79.50	756.2	37.0	3.79	0.020(2)	0.030(4)	1.5(1)

*The fragility of three CAS glasses were measured by isothermal equilibrium viscosity method in this study (see supplementary document), NS20 fragility is from Ref. [15].

Change 2 made in the manuscript: Section 3 has been added in the Supplementary document.

Supplementary Document

3. CAS fragility measurement by an isothermal equilibrium viscosity method

Equilibrium three-point beam-bending viscosity measurements were conducted for four charge-balanced CAS glasses $x\text{CaO}-x\text{Al}_2\text{O}_3-(1-2x)\text{SiO}_2$ ($x = 0.15, 0.2, 0.25, 0.3$) (the nomenclature of the CAS glasses is given by the SiO_2 content, i.e., the glass with $x = 0.15$ is designated as CAS70). Two of those glasses, CAS40 and CAS50, were measured twice for two separate sample preparations. The T_g -scaled Arrhenius plot of the viscosity of six measurements are shown in Fig. 3. Employing Angell's definition of the glass transition and fragility index, the fragility-index values were calculated and are listed in Table 1. The fragility-index error of CAS 60 and 70 glasses, which were measured once, is determined from the error of linear fitting of each Arrhenius plot. The errors of CAS 40 and 50, which have two measurements, are propagated from the errors of two independent linear fittings.

Fig. 3. T_g -scaled Arrhenius plot of the viscosity of four CAS glasses. Glasses CAS40 and CAS50 were measured twice for two individual sample preparations to test the measurement reproducibility.

Table 1 . Composition, T_g , fragility of four CAS glasses. The fragility error is derived from the error of the linear fits.

Glass ID.	Composition (mol%)			Mea.	T_g (K)	Fragility	
	SiO ₂	Al ₂ O ₃	CaO			Value	Error
CAS40	40.32	29.82	29.86	1 st	1127.0	53.9	0.6
				2 nd	1126.5	53.2	0.5
				Ave.	1126.7	53.6	0.8
CAS50	49.61	25.13	25.27	1 st	1132.8	49.9	0.9
				2 nd	1132.8	50.3	0.3
				Ave.	1132.8	50.1	0.9
CAS60	60.3	19.9	19.79	1 st	1137.4	43.9	0.4
CAS70	70.48	14.82	14.7	1 st	1148.1	40.5	0.6

4. The m value of the Fictitious high glass-forming liquid is assumed to be 70. But as far as I know, there are some oxide glasses that have m values much higher than 70. Thus, I think the assumption of m value of the Fictitious high glass-forming liquid is not reasonable.

Authors' reply: The Fictitious_{high} value of 70 is derived for the silicate glass with a ring structure; it does not apply to glasses with other structural types. We are not aware of any silicate glasses with $m > 70$. For clarification, we emphasize that the upper bound fragility of 70 is predicted only for the silicate glasses.

Changes made in the manuscript: A sentence is added after the Fictitious_{high} value discussion. (P15)

It is noted that this upper bound fragility value of 70 is predicted only for silicate glasses with a ring structure; glasses with other structural types, such as molecular materials and polymers, may have much higher fragility values, with an upper limit predicted to be 180 [37].

5. As for the MRD data of NAS glasses, they were obtained based on the assumption that the linear correlation between MRD and NBO/T is valid in the narrow composition range. However, there are only three data points in Fig. 4e, and the linear correlation between MRD and NBO/T is very poor. Therefore, I think that the MRD data of NAS glasses may have great inaccuracy.

Authors' reply: We share Reviewer #2's concern. There are only two lines (NAS65 and NAS70) made by four data points for NAS in a narrow NBO/T range. We have already acknowledged the possibility of such an inaccuracy in our manuscript as "The linear equation averaged from the two linear lines derived from 65 (solid blue) and 70 (dotted blue) mol% SiO₂ NAS glasses, as shown in Fig. 4e, was used to calculate the MRD-values of the m -measured NAS glasses, **based on the assumption** that the linear correlation between MRD and NBO/T ratio is valid in the narrow composition range of the 10 NAS glasses with constant SiO₂ mol% contents. This is just **a rough estimation to get approximate MRD values of NAS glasses for which neutron-scattering data were unavailable.**" Even with such a great inaccuracy, the inverse m -MRD correlation is qualitatively demonstrated, by comparing the changing trends of m in Fig. 4b with the changing trends of MRD in Fig. 4e. The higher m -values of CAS glasses in Fig. 4b are associated with their lower MRD-values in Fig. 4e. Moreover, the larger m -slope of NAS in Fig. 4b is in line with the larger negative MRD-slope in Fig. 4e.

Changes made in the manuscript: We rephrased the sentences in P12 to make the message clearer.

Linear trends of *MRD* vs. *NBO/T* ratio are assumed for both the 65 (solid blue) and 70 (dotted blue) mol% SiO₂ NAS glasses, as shown in Fig. 4e. The fitted line was subsequently used to estimate the *MRD*-values of the *m*-measured NAS glasses, based on the assumption that the linear correlation between *MRD* and *NBO/T* ratio is valid in the narrow composition range of the 10 NAS glasses reported (all at constant SiO₂ mol% content). This is admittedly a rough estimation to obtain approximate *MRD* values of NAS glasses for which neutron-scattering data were unavailable.

6. What's the physical meaning of slope ratio (liquid/glass) of *MRD*? Fragility is defined as the slope ratio of viscosity when temperature is approaching *T_g*. The slope ratio of *MRD* between liquid and glass is a dimensionless value, which cannot be related to fragility. Why did not the authors directly compare the slope of *MRD* around *T_g* with fragility?

Authors' reply: The slope ratio (liquid/glass) of *MRD*, $\alpha_{liq}/\alpha_{glass}$, is an alternative way to express the α_{glass} -scaled configurational CTE (α_{conf}). The latter is defined as the difference between macro-scale bulk-liquid and glass CTE values, $\alpha_{conf} = \alpha_{liq} - \alpha_{glass}$. By definition, $\alpha_{conf}/\alpha_{glass} = \alpha_{liq}/\alpha_{glass} - 1$.

A relationship between viscous dynamics and the bulk configurational CTE (α_{conf}) of silicate glasses has been discovered [37]; it illustrates the fact that “strong” glass-forming liquids show a small change in CTE at *T_g*, while “fragile” glass-forming liquids show a large change in CTE at *T_g*. That is the origin of the positive correlation between $\alpha_{liq}/\alpha_{glass}$ and *m* observed in Fig. 6d.

The above discussion, which is also presented in the manuscript, demonstrates that it is the slope ratio, $\alpha_{liq}/\alpha_{glass}$, that correlates with *m*. The slope of *MRD* around *T_g*, i.e. α_{liq} , is just the CTE of the supercooled liquid which includes both vibrational and configurational CTE, and can not be correlated with *m*.

Changes made in the manuscript: Based on the above discussion, no change is made in the manuscript.

7. Finally, there are some minor mistakes. Abbreviations of words like CAS, NAS, and CTE do not give the full names. There is a double “[3]” error on line 465.

Authors' reply: We thank the Reviewer #2 for his/her careful review. We added the full names of CAS, NAS and CTE and remove the double [3].

Based on the above considerations, I believe it is not suitable to consider to publish the manuscript in the NC journal.

Authors' reply: We thank Reviewer #2 for his/her thorough review and we hope that our clarifications and corrections have strengthened our manuscript and addressed Reviewer #2's questions and concerns so that the paper is now suitable for publication.

Reply to Reviewer #3's comments

Reviewer #3: This paper reports the relationship between the fragility (m) and the medium-range order in silicate glasses. The major experimental finding is the inverse linear relationship between the fragility and the average medium-range distance MRD deduced from the first sharp diffraction peak. This was based on ~80 data points from CAS, NAS, and other silicate glasses. The authors went on to argue that the MRD value is related to the excess of small network rings, which they claim are less stable, and later on discussed the results in the context of coarse-grained model (aka the linkage of A-O-A networks).

The paper is well-written and a pleasure to read. The experimental results are solid.

Authors' reply: We thank the Reviewer #3 for his/her understanding of, and encouraging comments on, our manuscript.

However, to me, the paper fell short of providing a convincing explanation for the observed correlation. It would be more effective if the authors have done some simulations, as they have in their earlier publications, to elucidate the origin of the m -vs-MRD correlation.

Authors' reply: We agree with Reviewer #3's comment that our findings could be strengthened by simulations. Here, to address this comment, we conducted some new simulations to demonstrate "how low MRD values are correlated with the excess amounts of small rings", as shown later (reply to comment 1.3).

Changes made in the manuscript: no changes were made in the manuscript regarding this comment.

The presented experimental results with heating are nice but they do not give definitive answers. After all, there are only three data points (Fig. 6).

Authors' reply: We have added another glass, NS20, which has been measured by *in-situ* neutron scattering and has a lower m value of 37. Now Fig. 6 (c)-(e) contains four data points to support the linear correlation. We hope that this addition strengthens the support for the inverse correlation of m with MRD.

Changes made in the manuscript: Glass NS20 was added in Fig. 6 (P18), and all the other related changes were made in the manuscript accordingly.

Fig. 4. Structural origin of the m -MRD correlation revealed by *in-situ* neutron total-scattering analysis. (a) FSDP regions of $F(Q)$ of three CAS glasses measured from room temperature up to $1.1 T_g$ and NS20 glass from RT to $1.25 T_g$. The RT- $F(Q)$ is plotted in black, then the color changes following the rainbow color spectrum, until the $F(Q)$ of $1.1 T_g$ is plotted in red. The three $F(Q)$ s with T higher than $1.1 T_g$ are plotted by gray dash lines. Both CAS40 and CAS50 glasses were measured at ten temperatures, as labeled in the figure legend; due to time limitation, CAS70 was only measured at seven temperatures with $T = 0.4, 0.6$ and $0.8 T_g$ skipped. (b) An analogous OLUA plot of scaled- MRD_T / MRD_{T_g} as a function of scaled temperature, T/T_g for the four glasses. The slopes for the structural change in both supercooled-liquid and glass ranges are determined by linear fits, as shown by the dotted lines. Note the significant discontinuous increase of MRD_T / MRD_{T_g} on passing through T_g . The inverse linear correlation between the through- T_g slope ratio (liquid/glass) of MRD as a function of T/T_g and RT-MRD (c), along with the positive correlation between the through- T_g slope ratio and fragility- m (d), provide a semi-quantitative justification for the inverse linear correlation between m and RT-MRD (e), as illustrated in **Error! Reference source not found.** for 88 glasses. The error of the slope ratio is propagated from the linear fitting error.

Simulations with a few model silicate systems, especially numerical results showing **how low MRD values are correlated with the excess amounts of small rings**, would significantly strengthen the

manuscript.

Authors' reply: We followed Reviewer #3's suggestion by conducting some new simulation to support out previous experimental results. Specifically, we simulated three CAS glasses and computed their ring-size distributions. The inverse correlation between *MRD* and relative percentage of ≤ 4 -membered rings (Fig. 7d) confirms that the *MRD* can be used as a parameter representing the percentage of small-sized rings.

Change 1 made in the manuscript: a paragraph and Fig. 7 is added in the section "*Physical interpretation of m-MRD correlations and beyond*". (P19-20)

Main text

Physical interpretation of m-MRD correlations and beyond. In the following, we offer a physical picture to explain the origin of the inverse linear correlation between the *MRD* value at room temperature and the slope ratio_{liquid/glass} of the *MRD* (and, hence, fragility). To confirm that the *MRD* can be used as a ring-size index, we first reveal the valid structure of three CAS glasses by combining neutron-diffraction experiments and force-enhanced atomic refinement (FEAR) simulations [12]. Fig. 5a presents structure factors $F(Q)$ for three CAS glasses, comparing the neutron-measured $F(Q)$ with simulations by FEAR and MD (for comparison purposes), along with a similar comparison for the $F(Q)$ of fused silica (FS) from Ref. [12]. The fact that the FEAR structure offers an excellent description of the FSDP of the experimental structure factor of all glasses (significantly improved as compared to MD), indicates that the FEAR model provides a realistic description of the glasses' medium-range order. We then computed the ring-size distribution from the FEAR-simulated structures using the RINGS code [40] by using the Guttman's ring definition. Each glass' ring-size distribution is individually plotted in Fig. 5b as a histogram of absolute ring counts within the simulation. In line with the experimental RingFSDP method, which groups the differently sized rings in three families [27], the grouped ring size distribution (i.e., ≤ 4 -, 5-, and ≥ 6 -membered rings) for all four glasses are compared in one plot (Fig. 5c). The population of small sized, ≤ 4 -membered rings, decreases significantly and systematically with increasing SiO_2 content, while, in turn, the population of intermediate 5-membered remain fairly stable and the population of large ≥ 6 -membered rings increase upon

increasing SiO₂ mol% content. The inverse correlation between *MRD* and the relative percentage of ≤ 4 -membered rings (Fig. 5d) confirms that the *MRD* can be used as a parameter that reflects the percentage of small-sized rings in a glass' structure. As such, low *MRD* values are indicative of the existence of higher amounts of small rings in the glass network. Detailed information about the FEAR simulations and ring-size distribution calculations is reported in Section 4 of the Supplementary Document.

Fig. 5 | FEAR simulation confirms that *MRD* can be used to represent the percentage of small-sized rings. **a** Measured (black dotted curves) and two simulated, MD (red solid curves) and FEAR (blue solid curves) total reduced structure factors $F(Q)$ for three CAS and FS glasses, with each glass' $F(Q)$ s vertically shifted up by 2 for clarity. **b** Guttman's ring-size distribution for each glass. **c** Comparison of grouped ring-size distribution for all glasses. **d** Inverse correlation between *MRD* and relative percentage of ≤ 4 -membered rings.

Change 2 made in the manuscript: Section 4 has been added in the Supplementary document.

Supplementary Document

4. FEAR simulation of CAS glasses and ring-size quantification

We simulated the structure of three CAS silicate glasses by combining neutron-diffraction experiments and force-enhanced atomic refinement (FEAR) [12]. All simulations were carried out using the Large-scale Atomic/Molecular Massively Parallel Simulator packages [2]. Three CAS glass models, CAS40, CAS50 and CAS70, were computed by molecular-dynamics (MD) simulations with each model comprising around 3000 atoms. We applied the interatomic potential parametrized by Jakse – as it has been found to yield some structural and elastic properties that are in good agreement with experimental data for CAS [3]. A cutoff distance of 8.0 Å was used for the short-range interactions. The Coulombic interactions were calculated by adopting the Fennell damped shifted force model with a damping parameter of 0.25 Å⁻¹ and a global cutoff of 8.0 Å. These three glasses were first simulated by MD simulations using a conventional melt-quench method, as described in the following. First, the atoms were randomly placed in a cubic box using PACKMOL [4] while ensuring the absence of any unrealistic overlap. The systems were then subjected to an energy minimization, followed by some 100 ps relaxations in the canonical (*NVT*) and isothermal-isobaric (*NPT*) ensembles at 300 K, sequentially. These models were then fully melted at 3000 K for 100 ps in the *NVT* and, subsequently, *NPT* ensemble to ensure the loss of the memory of the initial configurations and to equilibrate the system. Then these liquids were cooled from 3000 K to 300 K in the *NPT* ensemble at zero pressure with a cooling rate of 1K/ps. For all simulations, we adopted the Nosé-Hoover thermostat and a fixed time step of 1 fs.

We then assessed the ability of the FEAR [5] (force-enhanced atomic refinement) method to offer an improved description of the atomic structure of glassy silica as compared to those generated by MD or reverse Monte Carlo (RMC). To this end, we adopted the FEAR methodology introduced by Drabold et al. [6]. In contrast to MD simulations (which solely uses the knowledge of the interatomic potential) and RMC [7] simulations (which solely uses the knowledge of experimental data), the FEAR approach leverages all the available information. FEAR presents

two key advantages: (i) it is more computationally efficient, since the energy does not need to be computed at every RMC step, and (ii) it does not rely on any assumption regarding the weights associated with the structural and energy terms in the cost function. In detail, we first started from a “randomized” structure generated by RMC while using a very high effective temperature, namely, $T_{\chi} = 5000$ K. Following the original implementation of the FEAR method, the system was then iteratively subjected to a combination of RMC refinements and energy-minimization steps, wherein each FEAR iteration consists of: (i) 3600 RMC steps and (ii) an energy minimization (conducted with the conjugate- gradient method). We found that 16 of such iterations were sufficient to achieve a convergence of the potential energy and R_{χ} for the CAS glasses. During the refinement, we dynamically adjusted the average acceptance probability of the Metropolis algorithm by linearly decreasing the effective temperature T_{χ} from 10^2 down to 10^{-3} during the FEAR refinement. These parameters were found to yield a glass structure exhibiting minimum R_{χ} and potential-energy values.

We then explored the structure of the glass structures generated by FEAR. To this end, we computed the neutron structure factor for each of the simulated glasses. Fig. 6 shows the reduced structure factor, $F(Q)$, predicted by FEAR and MD, which are compared with experimental neutron-diffraction data. We observe that the FEAR-derived structure factors exhibit an excellent agreement with the experimental data over the entire Q range—which is not unexpected, since the neutron PDFs were used as input for the FEAR simulations. In contrast, the MD-derived structure factors present some notable discrepancies with the experimental data. Especially, FEAR predicts a sharper FSDP than MD, suggesting that the glass refined using FEAR exhibits a more ordered medium-range structure than its MD-based counterpart. This echoes the fact that glass structures generated by FEAR tend to exhibit an increased thermodynamic stability (i.e., lower energy) as compared to structures generated by MD.

Fig. 6 | Measured (black dotted curves) and two simulated, MD (red solid curves) and FEAR (blue solid curves), total reduced structure factors $F(Q)$ for three CAS glasses. Each glass $F(Q)$ is vertically shifted up by 2 for clarity.

We then computed the ring-size distribution from the FEAR-simulated structures using the RINGS code [8]. The Guttman's ring definition was used for ring-counting since it is the most relevant to describe the ring distribution derived from the FSDP of scattering patterns in terms of the probed length scale [12]. The relative ring-size distributions derived from the direct ring counting of the FEAR structure models were compared with the RingFSDP results and are shown in Fig. 7. There is a significant difference for the percentage of large-sized, ≥ 6 -membered rings between the FEAR and RingFSDP methods. Especially for CAS40 and CAS50 glasses, about 35% of large ≥ 6 -membered rings are counted from the FEAR model while RingFSDP shows no large-sized rings. However, both methods show a similar trend, namely, the fraction of small-

sized, ≤ 4 -membered rings significantly and systematically decreases with increasing SiO_2 content.

Fig. 7 | Ring-size distribution comparison between the experimental RingFSDP and FEAR simulation analysis.

Some other comments.

1. As MRD is central to the discussion, it would be helpful to the readers to outline in the main text how MRD is determined (Eq. (2-3) in the Methods) and how it is treated as a length-weighted ring-size index.

Authors' reply: We thank Reviewer #3' for his/her comment and we introduced the definition of MRD and its determination in P6 of the main text.

Changes made in the manuscript: MRD definition and calculation are moved to P6 from the Method section.

Specifically, we use the recently developed RingFSDP method [27] to characterize the medium-range order structure of these glasses (see Methods). The reciprocal-space FSDP can be

deconvolved into three modified Gaussian peaks with fixed positions, which correspond to real-space distances of 3.15, 3.70 and 4.30 Å, respectively. Each peak is ascribed to a certain family of rings: (i) large rings (≥ 6 -membered) centered at low Q ; (ii) medium rings (5-membered) centered at intermediate Q ; and (iii) small rings (≤ 4 -membered) centered at large Q . The integrated area of each peak is proportional to the absolute number of such specified size rings. The relative ring-size distribution (f_n) is calculated from the ratio of the integrated peak area ($I_{S_n(Q)}$) to the total FSDP area ($I_{S_{\text{FSDP}}(Q)}$) using:

$$f_n = \frac{I_{S_n(Q)}}{I_{S_{\text{FSDP}}(Q)}}. \quad (1)$$

Then, the average medium-range distance (MRD) can be calculated as:

$$MRD (\text{Å}) = f_{\leq 4\text{ring}} \times 3.15 + f_{5\text{ring}} \times 3.70 + f_{\geq 6\text{ring}} \times 4.30 \quad (2)$$

The MRD structural parameter, which represents an average medium-range distance of the glass structure, can also be treated as a length-weighted ring-size index. The smaller the value of MRD , the greater is the number of small-size rings. It is reciprocally related to the average position of the FSDP.

2. Is there an estimate of the energy barrier for deforming the small rings (say at room temperature)?

Authors' reply: We do not have an estimation of the absolute energy for deforming small rings, but we estimate that it takes 4.3 eV less energy to deform a 4-membered ring than a 5- or 6-membered ring, as explained below.

Changes made in the manuscript: A new paragraph is added in the Section of “*Physical interpretation of m -MRD correlations and beyond*”. (P22)

In Ref. [32], the relative energies of different n -membered rings are first defined as $\Delta E_n/E_0$, with E_0 being an energy scale related to the cohesive energy, which is derived as 1744 kJ/mole for FS [46], corresponding to 18 eV per SiO_2 molecule. The $\Delta E_n/E_0$ values were then calculated by examining the interatomic distances and bond-angle distributions in n -membered rings derived

from an MD-simulated model of FS. In Fig.12 of Ref. [32], it is shown that both 5- and 6-membered rings have close to a zero energy, while the 4-membered rings have an estimated $\Delta E_4/E_0$ value of 0.06, which corresponds to a penalty energy of 4.3 eV for a 4-membered ring composed of 4 SiO₂ molecules. This shows that the energy of small-sized 4-membered rings is 4.3 eV higher than those of 5- and 6-membered rings. Therefore, we connect the relative energy of n -membered rings with the potential parameter, λ ; the higher the relative energy of a ring, the higher the steepness of λ , the higher the fragility. Vice versa, a glass with a higher population of large-sized rings has a lower energy and exhibits a lower fragility, which echoes the finding that “*soft colloids make strong glasses*” [47].

REVIEWERS' COMMENTS

Reviewer #1 (Remarks to the Author):

The authors improved the ms due to the comments of the reviewers very much. So in general I am happy with the state of the ms. But if one reads the abstract and the summary one could get the impression the findings of the authors mirrors the universal behaviour of glassy materials. This is not the case here. Therefore I agree with publication but only after mayor revision of the "universal statements" in the ms.

Reviewer #2 (Remarks to the Author):

In the revised version, and I found that all my concerns in the original MS has been clearly clarified. It provides a new evidence of the correlation between liquid fragility and local structure. I feel happy to recommend the paper in the revised version to publish in the Journal of Nature Communications.

Reviewer #3 (Remarks to the Author):

The authors have gone extra lengths to address the referees' comments. I particularly appreciate the new simulation results and the added data point.

For the last question about the energy barrier for deforming the small rings, the authors gave an estimate of 4.3 eV. Looking at their answers, I believe that this is the bond-breaking energy, rather than for deforming the rings. In this light, I recommend that the authors remove the newly added paragraph on the estimation of energy barriers.

Other than this point, I recommend acceptance of the manuscript.

We would like to thank all the reviewers again for their second-round reviews and helpful comments, which clearly resulted in a strengthened paper.

Below are our point-by-point replies to the comments raised by the reviewers, including details on how we have revised the manuscript.

For clarification, the format is defined below:

Reviewers' comments (gray highlighted)

Authors' replies (blue font)

Changes made in the manuscript (black Arial font with double line spacing)

Reply to Reviewer #1's comments

Reviewer #1: The authors improved the ms due to the comments of the reviewers very much. So in general I am happy with the state of the ms. But if one reads the abstract and the summary one could get the impression the findings of the authors mirrors the universal behaviour of glassy materials. This is not the case here. Therefore I agree with publication but only after mayor revision of the "universal statements" in the ms.

Authors' reply: We thank the Reviewer #1 for the acknowledgement. In our manuscript we clearly state that we do not intend to propose a universal structural descriptor for fragility, because we believe that glasses with very different structures (e.g., non-network glasses) should exhibit different structure-dynamics mechanisms. Moreover, we doubt the existence of a universal rule that is generically applicable for all structural types of glassy materials. However, we used the word "universal" four times in P14-15 to illustrate the *m-MRD correlation for 88 glasses*. To avoid the confusion, we remove all these "universal" words.

Changes made in the manuscript: we remove all the four "universal" words, three in the second paragraph of P14, one is in the Figure caption of Fig. 5.

Reviewer #2: In the revised version, and I found that all my concerns in the original MS has been clearly clarified. It provides a new evidence of the correlation between liquid fragility and local structure. I feel happy to recommend the paper in the revised version to publish in the Journal of Nature Communications.

Authors' reply: We thank the Reviewer #2 for the acknowledgement.

Reviewer #3: The authors have gone extra lengths to address the referees' comments. I particularly appreciate the new simulation results and the added data point.

For the last question about the energy barrier for deforming the small rings, the authors gave an estimate of 4.3 eV. Looking at their answers, I believe that this is the bond-breaking energy, rather than for deforming the rings. In this light, I recommend that the authors remove the newly added paragraph on the estimation of energy barriers.

Other than this point, I recommend acceptance of the manuscript.

Authors' reply: We thank the Reviewer #3 for this acknowledgement. We agree with the Reviewer #3 that we should not use the bond-breaking energy to estimate the ring deformation energy, we follow the Reviewer's suggestion to remove the part of energy barrier estimation and only qualitatively state that the energy of 4-membered rings is higher than those of 5- and 6-membered rings.

Changes made in the manuscript: we remove the energy estimation part and rewrite the paragraph as shown below (P22).

In Ref. [32], the relative energies of different n -membered rings are first defined as $\Delta E_n/E_0$, with E_0 being an energy scale related to the cohesive energy. The $\Delta E_n/E_0$ values were then calculated by examining the interatomic distances and bond-angle distributions in n -membered rings derived from an MD-simulated model of FS. In Fig.12 of Ref. [32], it is shown that both 5- and 6-membered rings have close to a zero energy, while the 4-membered rings have an estimated $\Delta E_4/E_0$ value of 0.06. This shows that the energy of small-sized 4-membered rings is higher than those of 5- and 6-membered rings.